# Modifications at K31 on the lateral surface of histone H4 contribute to genome structure and expression in apicomplexan parasites

Fabien Sindikubwabo[1], Shuai Ding[2], Tahir Hussain[1], Philippe Ortet[3], Mohamed Barakat[3], Sebastian Baumgarten[2], Dominique Cannella[1], Andrés Palencia[1], Alexandre Bougdour[1], Lucid Belmudes[4], Yohann Couté[4], Isabelle Tardieux[5], Cyrille Y Botté[6], Artur Scherf[2], Mohamed-ali Hakimi[1]*

[1]Institute for Advanced Biosciences (IAB), Team Host-pathogen interactions and immunity to infection, INSERM U1209, CNRS UMR5309, Université Grenoble Alpes, Grenoble, France; [2]Unité de Biologie des Interactions Hôte-Parasite, Institut Pasteur, CNRS, ERL 9195, INSERM, Unit U1201, Paris, France; [3]Aix-Marseille Univ, CEA, CNRS, UMR 7265, BIAM-LEMIRE, St-Paul-lez-Durance, France; [4]Université Grenoble Alpes, CEA, INSERM, Grenoble, France; [5]Institute for Advanced Biosciences (IAB), Team Membrane and Cell Dynamics of Host Parasite Interactions, INSERM U1209, CNRS UMR5309, Université Grenoble Alpes, Grenoble, France; [6]Institute for Advanced Biosciences (IAB), Team ApicoLipid, INSERM U1209, CNRS UMR5309, Université Grenoble Alpes, Grenoble, France

*For correspondence:
mohamed-ali.hakimi@univ-grenoble-alpes.fr

Competing interests: The authors declare that no competing interests exist.

**Abstract** An unusual genome architecture characterizes the two related human parasitic pathogens *Plasmodium falciparum* and *Toxoplasma gondii*. A major fraction of the bulk parasite genome is packaged as transcriptionally permissive euchromatin with few loci embedded in silenced heterochromatin. Primary chromatin shapers include histone modifications at the nucleosome lateral surface close to the DNA but their mode of action remains unclear. We now identify versatile modifications at Lys31 within the globular domain of histone H4 that crucially determine genome organization and expression in Apicomplexa parasites. H4K31 acetylation at the promoter correlates with, and perhaps directly regulates, gene expression in both parasites. By contrast, monomethylated H4K31 is enriched in the core body of *T. gondii* active genes but inversely correlates with transcription, whereas it is unexpectedly enriched at transcriptionally inactive pericentromeric heterochromatin in *P. falciparum*, a region devoid of the characteristic H3K9me3 histone mark and its downstream effector HP1.
DOI: https://doi.org/10.7554/eLife.29391.001

## Introduction

The *Apicomplexa* phylum clusters thousands of single-celled eukaryotes identified as parasites of metazoans including humans in whom they cause or increase the risk of major public health problems. Preeminent human pathogens include *Plasmodium* species that are responsible for dreadful malarial disease, as well as *Toxoplasma gondii* and *Cryptosporidium* spp., which are leading causes of food-borne and water-borne diseases. A shared characteristic of apicomplexan life cycles is the multiplicity of developmental stages, with progress from one stage to the next occurring alongside precise genetic reprogramming to ensure the survival and transmission of parasite populations. The

emerging concept of remarkably dynamic gene expression in Apicomplexa has risen from the observation that large numbers of mRNAs are exclusively expressed in a given developmental stage (*Bozdech et al., 2003*; *Radke et al., 2005*).

Unlike those of metazoans, Apicomplexa genomes have a unique chromatin architecture typified by an unusually high proportion of euchromatin and only a few heterochromatic islands, which are scattered through the chromosome bodies or embedded at telomeres and centromeres. Although alterations in chromatin structure are acknowledged to be important for the transcriptional control of commitment to stage differentiation in several Apicomplexa, as well as for antigenic variation-mediated immune evasion in *P. falciparum*, yet the understanding of chromatin remodeling remains incomplete (*Bougdour et al., 2010*; *Scherf et al., 2008*).

In eukaryotes, the timely opening and closing of chromatin that is required for gene expression, chromosomal organization, DNA repair or DNA replication is governed by histone turnover and the post-translational modifications (PTMs) of histones, such as lysine methylation (me) and acetylation (ac) among many others. Indeed, PTMs on histone tails have been shown, alone or in combination, to alter the accessibility of effector proteins to nucleosomal DNA and hence to impact chromatin structure, according to the 'histone code' hypothesis (*Strahl and Allis, 2000*; *Turner, 2000*). In addition, PTMs also act as signals that recruit ATP-dependent remodeling enzymes to move, eject or reposition nucleosomes. Enzymes carrying antagonist activities account for the versatile activities of PTMs on chromatin: it is the opposite, yet well concerted, activities of histone acetyltransferases (HATs) and histone deacetylases (HDACs) that control acetylation levels in cells.

Apicomplexa have evolved unique ways to modify histones that rival the strategies adopted by the cells they infect and provide zoites with remarkable capacities to multiply or undergo stage differentiation (*Bougdour et al., 2010*). Some histone-modifying enzymes have acquired gain-of-function mutations that confer broader or enhanced activity on substrates. It is the case of parasite Set8-related proteins endowed with H4K20 mono-, di-, and trimethylase activities that contrast with the mono-methylase-restricted metazoan Set8, and that derive from a single-amino-acid change in the substrate-specific channel (*Sautel et al., 2007*). Another example is the Apicomplexa HDAC3 family which is typified by an AT motif insertion at the entrance of the active-site tunnel of the conserved catalytic domain causing additional substrate/inhibitor recognition and binding properties (*Bougdour et al., 2009*).

In eukaryotes, PTMs have been detected primarily in the histone tails sticking out from the nucleosome, but an ever-growing list of PTMs characterized as critical regulators of chromatin structure and function have now been identified in the lateral surface of the histone octamer, which directly contacts DNA (*Lawrence et al., 2016*; *Tropberger and Schneider, 2013*). Those 'core' histone PTMs promote different outcomes in nucleosome dynamics depending on their location. Modifications near the DNA entry-exit region of the nucleosome (e.g., H3K36ac) have been shown to favor local unwrapping of DNA from a histone octamer, thereby providing a better exposure of nucleosomal DNA to chromatin-remodeling and DNA-binding proteins (*Neumann et al., 2009*). On the other hand, lateral-surface PTMs mapping close to the dyad axis (e.g., H3K122ac) were shown to decrease the affinity of the octamer to DNA and to affect nucleosome stability or mobility significantly (*Tropberger et al., 2013*). As has been described for histone-tail modifications, different lateral-surface modifications on the same residue can be associated with opposite transcriptional programs. This is the case for the H3K64 residue near the dyad axis that facilitates nucleosome eviction and thereby gene expression when acetylated (*Di Cerbo et al., 2014*), whereas trimethylation of the same residue acts as a repressive heterochromatic mark (*Daujat et al., 2009*).

In both *T. gondii* and *P. falciparum*, unbiased mass spectrometry hasuncovered the repertoire of the most prevalent histone PTMs, including singular marks. However, only few of them were mapped at the outer surface of the octamer (*Saraf et al., 2016*; *Trelle et al., 2009*). In this study, we investigated in depth how histone H4 PTMs could influence chromosome organization and gene regulation in apicomplexan parasites. We report versatile modifications at lysine 31 of histone H4 (H4K31), which lies at the protein–DNA interface close to the dyad axis of the nucleosome. Genome-wide mapping revealed that H4K31 could be either acetylated or methylated and that enrichment of these marks occurred in a mutually exclusive manner. In *T. gondii*, the H4K31 residue tended to be acetylated at the promoter of a nearby active gene and to be mono-methylated in the core body of the gene. H4K31me1 occupancy was inversely correlated with gene expression, suggesting that the mark acts as a repressive mark that impedes RNA polymerase progression. In *P.*

*falciparum*, H4K31ac was also seen at the promoter whereas H4K31me1 occupancy was highly enriched at pericentromeric heterochromatin, possibly compensating for the absence of H3K9me3 and HP1 in this atypical chromosome structure in order to maintain a constitutive heterochromatin environment.

## Results

### H4K31 maps at the dyad axis of the nucleosome

While studying the protein content of nucleosomes from *T. gondii*-infected cells, we and others mapped an acetylation site on histone H4 lysine 31 (H4K31ac) that had previously been largely underestimated (*Figure 1a*) (*Jeffers and Sullivan, 2012*; *Xue et al., 2013*). This modification was also identified in both the proteome (*Saraf et al., 2016*) and acetylome (*Cobbold et al., 2016*) of *P. falciparum* throughout the intraerythrocytic developmental cycle of the parasite. The H4K31 residue is located at the N-terminus of the H4 α1 helix and its side chain is extended in the major groove of the DNA (*Figure 1b and c*). The closed state of chromatin results from the interaction of K31 and R35 residues with DNA by a water-mediated hydrogen bond. The addition of an acetyl group to the -NH2 group of the lysine side chain (K31) abolishes its interaction with DNA mediated by a water molecule (*Figure 1c*, right panel). Acetylation may therefore destabilize the protein–DNA interface close to the dyad axis of the nucleosome where the residue lies, and thus presumably could open the chromatin. Although the H4K31 residue is well conserved across species (*Figure 1b*), mass spectrometry initially indicates that its acetylation is restricted to metazoans, as an unexpected mark in inflammatory and auto-immune contexts (*Garcia et al., 2005*; *Soldi et al., 2014*), as it was found neither in yeast nor in the ciliated protozoan *Tetrahymena* (*Garcia et al., 2007*). Recent studies, including our present findings, contradict this view as they show that this PTM also arises in the Apicomplexa phylum (*Cobbold et al., 2016*; *Jeffers and Sullivan, 2012*; *Saraf et al., 2016*).

### H4K31ac marks euchromatin in mammalian cells and apicomplexan parasites

Although H4K31ac was unequivocally identified by mass spectrometry in both *T. gondii* (*Jeffers and Sullivan, 2012*; *Xue et al., 2013*) and *P. falciparum* (*Cobbold et al., 2016*; *Saraf et al., 2016*) (*Figure 1a*), the dynamics and nuclear distribution of the mark during infections remain understudied. To further probe in situ the kinetics of this histone mark in apicomplexans, we raised an antibody against a synthetic peptide that is acetylated at the H4K31 position whose specificity was controlled by dot-blot assays. First, no cross-reactivity with the unmodified peptides (*Figure 1—figure supplement 1A*) or with previously described acetyl and methyl marks in histone tails and globular domains (*Figure 1—figure supplement 1*) was detected. Second, H3K14ac-directed antibodies did not cross-react with H4K31 peptides (*Figure 1—figure supplement 1A*) whereas they properly detected H3K14ac-containing peptides (*Figure 1—figure supplement 2*). Using human primary fibroblasts infected by *T. gondii*, we found H4K31ac to be distributed exclusively and uniformly within the nuclei of both parasite and human cells (*Figure 1d*). We also found that exposing cells to a histone deacetylase inhibitor (HDACi), e.g., FR235222, significantly increased H4K31ac signal intensity, which is otherwise moderate in parasite nucleosomes (*Figure 1e*). This is further evidence of the specific detection of acetylation. Similarly, response to HDACi treatment was observed by immunofluorescence analysis (*Figure 2*). Interestingly, PTMs at histone H3 tails, for example at H3K14ac and H3K27ac, were not altered upon FR235222 treatment under our conditions, which contrasts with the increased signal of H4K31ac and further confirms the specificity of the H4K31ac-directed antibodies (*Figure 2b*).

To gain insight into the behavior of H4K31ac during the *P. falciparum* intraerythrocytic developmental cycle (IDC), immunofluorescence assays were conducted over 48 hr of culture to probe the ring, trophozoite and schizont stages (*Figure 3a*). H4K31ac was typified by a nuclear signal throughout the IDC that increased upon HDACi treatment (*Figure 3b and c*). Overall, H4K31ac showed a nuclear punctate pattern, reminiscent of active loci clusters in specialized 'transcription factories' (*Figure 3b*) (*Mancio-Silva and Scherf, 2013*).

Strikingly, H4K31ac has remained understudied in other eukaryotes. To gain better resolution when imaging any of the nuclear or chromatin structures with which H4K31ac might be associated,

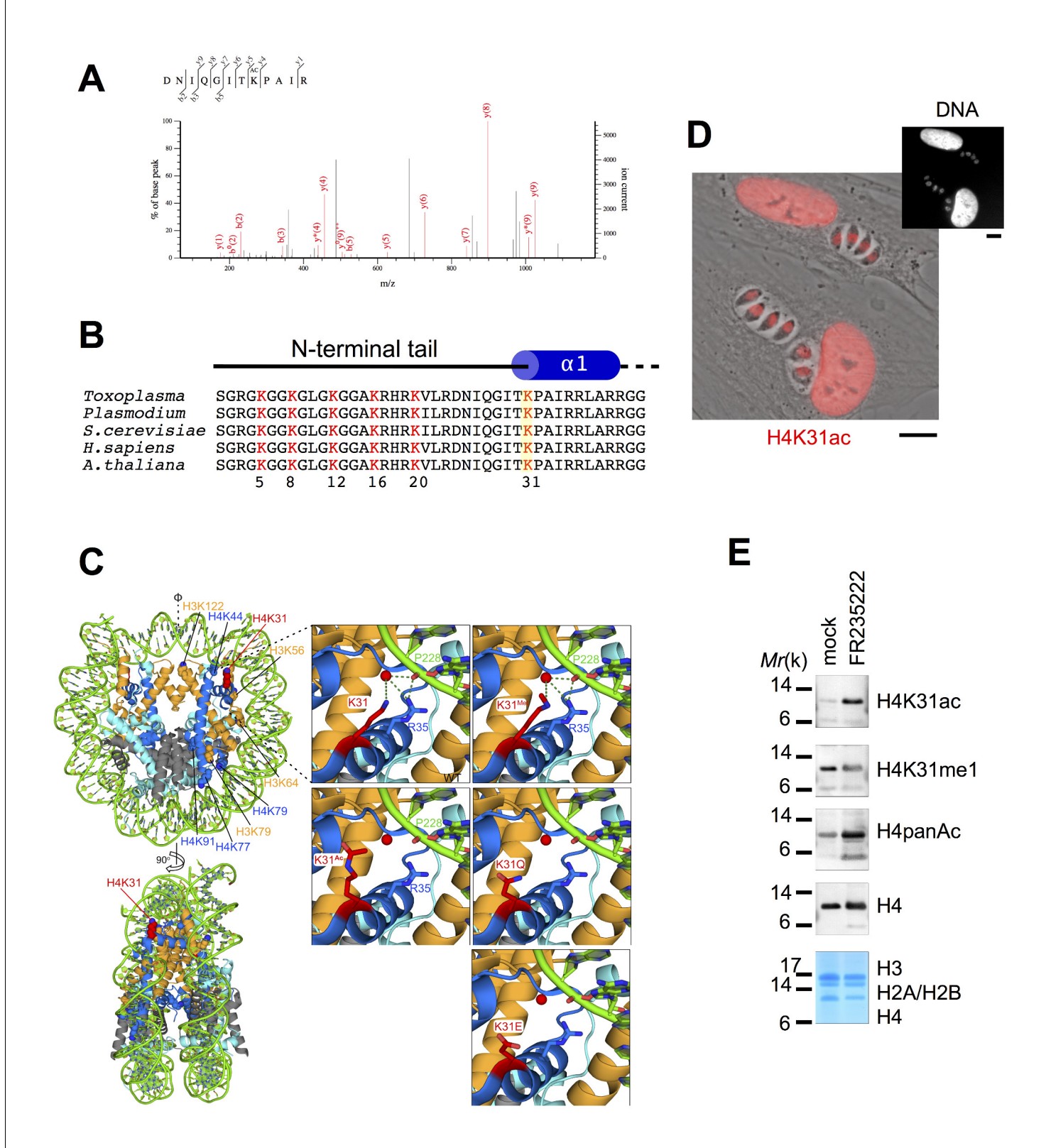

**Figure 1.** The residue K31 on the lateral surface of histone H4 is a novel PTM. (a) The high-resolution MS/MS spectrum of the H4K31ac peptide generated from histone H4. H4K31ac was identified using the Mascot search engine in the DNIQGITK(ac)PAIR peptide. (b) Sequence alignment of the

*Figure 1 continued on next page*

*Figure 1 continued*

first 42 residues of histone H4 from the indicated organisms. Yellow boxes highlight the conserved residue H4K31. (c) Structural analysis of H4K31 modifications. Ball-and-sphere representation of the nucleosome core particle, showing key H3 and H4 lysine residues that are known to be modified. The histone proteins of the nucleosome (PDB code: 3AFA) are color-coded as follows: H2A cyan, H2B grey, H3 orange and H4 blue. The H4K31 residue, highlighted in red, is placed at the dyad axis and mediates key interactions with the DNA (in green). The bottom panel is rotated 90 degrees around the molecular dyad axis. On the right, close-up diagrams of the interactions established by H4K31 with a water molecule (red sphere) and residue R35; and impact of the modifications: methylation, acetylation and succinylation (mimicked by mutant K31E). The mutant H4K31Q (PDB code: 3AZI) partially mimics lysine acetylation. (d) Immunofluorescence analysis of H4K31ac (in red) in both human foreskin fibroblast (HFF) cells and parasite nuclei. DNA (top inset) was stained with Hoechst. Scale bar, 10 μm. (e) Immunoblots of native purified nucleosomes from *T. gondii* parasites treated with FR235222 or DMSO for 18 hr. Data are representative of two independent experiments.

DOI: https://doi.org/10.7554/eLife.29391.002

The following figure supplements are available for figure 1:

**Figure supplement 1.** Specific binding of homemade H4K31ac-directed antibody to H4K31 acetylated peptide in vitro.

DOI: https://doi.org/10.7554/eLife.29391.003

**Figure supplement 2.** Specific binding of H3K14ac-directed antibody to H3K14ac-containing peptides in vitro.

DOI: https://doi.org/10.7554/eLife.29391.004

**Figure supplement 3.** Specific binding of homemade H4K31me1-directed antibody to H4K31 methylated peptide in vitro.

DOI: https://doi.org/10.7554/eLife.29391.005

**Figure supplement 4.** Specific binding ofH4K20me1-directed antibody to H4K20me1-containing peptides in vitro.

DOI: https://doi.org/10.7554/eLife.29391.006

we co-stained murine embryonic fibroblasts (MEFs) for DNA and various chromatin marks. H4K31ac was observed scattered through the nucleoplasm of MEFs but excluded from nucleoli and segregated away from heterochromatic foci, as were the transcription-associated PTMs H3K4ac, H3K9ac and H3K27ac (*Figure 3d*). This pattern is typically euchromatic and opposed to that revealed by the repressive marks H3K9me3 and H4K20me3, being associated instead with regions of highly condensed pericentromeric heterochromatin (*Figure 3e*). Taken together, these experiments show that H4K31ac displays an euchromatic pattern in both metazoans and apicomplexans.

## GCN5b and HDAC3 enzymes fine-tune the H4K31ac levels in *T. gondii*

To our knowledge, the enzymes that acetylate and deacetylate H4K31 remain unknown. FR235222 treatment induced a 5.3-fold increase in the H4K31ac signal in *T. gondii* nuclei when compared to that of other acetylated residues of the histone tails (*Figure 2b*), an observation that prompted us to look for potential deacetylases that target H4K31ac. To this end, we selected seven HDACi that cover the entire selectivity range for class I and II deacetylases, albeit with varying specificity profiles against apicomplexan parasites, as determined previously (*Bougdour et al., 2009*). We also included a specific inhibitor of protein translation in Apicomplexan halofuginone as a control (*Jain et al., 2015*). We found that cyclopeptide HDACi strongly enhanced H4K31ac levels in parasite nuclei, whereas other inhibitors had no effect (*Figure 4a and b*). These results are consistent with those showing that distinct point mutations at a single locus in an apicomplexan conserved region of TgHDAC3 abolishes the enzyme sensitivity to the cyclic tetrapeptide compounds (*Bougdour et al., 2009*). *T. gondii* is particularly suited to a single-gene perturbation strategy because its genome does not contain extensive HAT and HDAC paralogs, unlike mammalian genomes. To identify which of the five class-I/II HDAC homologues in *T. gondii* may account for the deacetylation of H4K31ac, we systematically performed CRISPR-mediated gene disruption. Inactivation of *TgHDAC3* but not of other *TgHDAC* genes caused hyperacetylation of H4K31 in parasite nuclei (*Figure 5a,b and c*), thereby mimicking the effect of the cyclic tetrapeptide HDACi on the enzyme (*Figure 4*) (*Bougdour et al., 2009*).

Reciprocally, we next sought the one or more responsible HATs targeting H4K31ac. We used a candidate-based approach by systematically depleting the parasite for key members of the three main HAT classes. Apicomplexans possess homologues of the Type A GCN5 and MYST family nuclear HATs as well as the Type B cytoplasmic HAT1 (*Vanagas et al., 2012*), but lack the PCAF (p300/CBP-associating factor) family, which is restricted to mammals. Intriguingly, *T. gondii* is unique among fellow apicomplexan parasites and other invertebrates in possessing two GCN5 HATs, designated TgGCN5a and TgGCN5b, that exhibit different histone acetylation activities (*Vanagas et al.,*

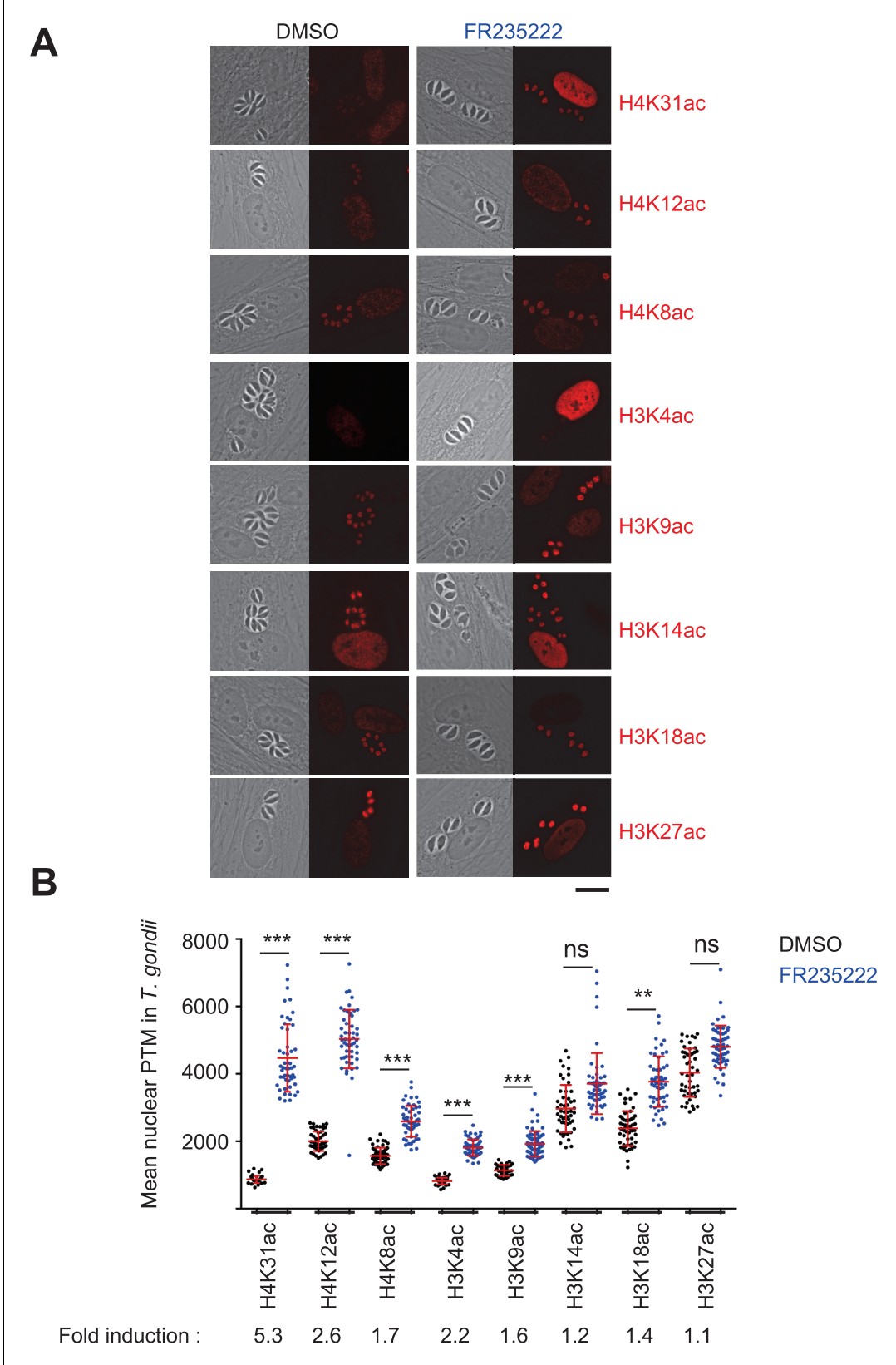

**Figure 2.** Immunofluorescence analysis of histone modifications in human cells infected with *T. gondii*. (a) Confluent monolayers of HFF cells were infected with tachyzoites in the presence of FR235222 and DMSO as a control. Immunofluorescence analyses (IFAs) were carried out with antibodies

*Figure 2 continued on next page*

*Figure 2 continued*

against specific acetylated histone H3 and H4 lysine residues as indicated. All modifications showed specific and distinct localizations in both parasite and host cell nuclei (in red). Scale bar, 20 µm. (b) Quantification of the intensity of the aforementioned PTMs staining in each parasite nucleus following FR235222 stimulation. Each symbol marks the PTM density of a single parasite nucleus. The results are represented as mean ± standard deviations from two independent experiments; the number of nuclei quantified was at least 50. Asterisks indicate statistical significance when comparing each individual FR235222-treated PTM sample and the corresponding control (DMSO) as determined by an unpaired two-tailed Student's t-test, **$p < 0.05$ and ***$p < 0.0001$; n.s., not significant. Fold induction of each PTM by FR235222 in parasite nuclei is indicated.

DOI: https://doi.org/10.7554/eLife.29391.007

*2012*). We first thought of *T. gondii* HAT1 as a promising candidate enzyme as the human HAT4 was shown to acetylate core PTMs in vivo, and even acetylated H4K31 although under in vitro conditions (*Yang et al., 2011*). However, cas9-mediated gene disruption of HAT1 had no effect on H4K31 acetylation (*Figure 6a and b*), and *TgGCN5b* was the only HAT-encoding gene whose disruption resulted in a drastic drop in H4K31ac signals in the parasite nuclei (*Figure 6a and b*). TgGCN5b is the prototypical GCN5 HAT in *T. gondii* because it can target H3K9, H3K14 and H3K18 (*Bhatti et al., 2006*). Furthermore, we noticed that the amino acid sequence surrounding H4K31 was not homologous to the preferred GCN5 consensus sites of acetylation found at H3K14 (*Kuo et al., 1996*) or H3K36 (*Morris et al., 2007*), yet the depletion of *TgGCN5b* led to a reduction of both H3K14ac (*Figure 6c and d*) and H4K31ac (*Figure 6a and b*) signals in vivo, suggesting that the repertoire of the lysine residues being acetylated by the GCN5 family is more diverse in *T. gondii* than in other organisms. Altogether these data clearly identify TgGCN5b as an H4K31 acetyltransferase whose activity is counteracted by TgHDAC3.

## H4K31me1 associates in vivo with distinct chromatin patterns

It is well appreciated that the targeting of lysine residues by acetylation, methylation or other PTMs cannot occur simultaneously. H4K31 has been identified as succinylated by proteomic mapping in *Drosophila*, yeast, and mammalian cells (*Xie et al., 2012*), as well as in *T. gondii* (*Nardelli et al., 2013*). Formylation is another modification targeting H4K31 in both human (*Wisniewski et al., 2008*) and Apicomplexan (*Nardelli et al., 2013*; *Saraf et al., 2016*) cells. While mass spectrometry-based proteomics strategies have allowed the characterization of H4K31 as a site of monomethylation in plant, budding yeast and metazoan cells, they failed to detect the PTM in *Tetrahymena* (*Garcia et al., 2007*; *Moraes et al., 2015*) and Apicomplexa (*Nardelli et al., 2013*; *Saraf et al., 2016*). By contrast, H4K31me2 has been detected in the *T. gondii* proteome (*Nardelli et al., 2013*). Aside from proteomic approaches, the possibility of 'dual' modifications on H4K31 has not been yet explored in apicomplexans or in any other species. Therefore, we raised antibodies against synthetic peptides containing mono- and di-methylated H4K31 and controlled for their specificity. The H4K31me2-directed antibodies, although reacting avidly with the peptide antigen, did not detect histone H4 in human or parasite protein extracts (data not shown). On the other hand, antibodies raised against H4K31me1-containing peptide nicely recognized histone H4 in purified *T. gondii* core histone extract (*Figure 1e*). The antibody is highly specific for synthetic H4K31me1 peptide (*Figure 1—figure supplement 3a*) over peptides with previously described acetyl and methyl marks in histone tails and globular domains in dot-blot assays (*Figure 1—figure supplement 3b,c and d*). As a control, H4K20me1-directed antibodies (*Sautel et al., 2007*) did not cross-react with H4K31 peptides (*Figure 1—figure supplement 3a*) while properly detecting H4K20me1-containing peptides in dot-blot assay (*Figure 1—figure supplement 4*). Taken together, these data show that our home-made antibodies are specific for the H4K31me1 epitope.

In situ, the H4K31me1 modification appeared to be uniformly distributed within the nucleus of dividing parasites, but surprisingly no signal was detected in the nucleus of the infected human cell (*Figure 7a*), although this PTM had been previously detected by mass spectrometry in human samples (*Garcia et al., 2007*). H4K31me1 was not (or barely) detected in interphase nuclei of either quiescent infected HFFs (*Figure 7a*) or uninfected MEFs (*Figure 7b*), but it decorated mitotic chromosomes , providing even more pronounced signals in the chromosome arms than the usual mitotic marker H3S10 phosphorylation (*Figure 7b*). These observations argue for a possible role for H4K31 methylation during cell division in mammalian cells.

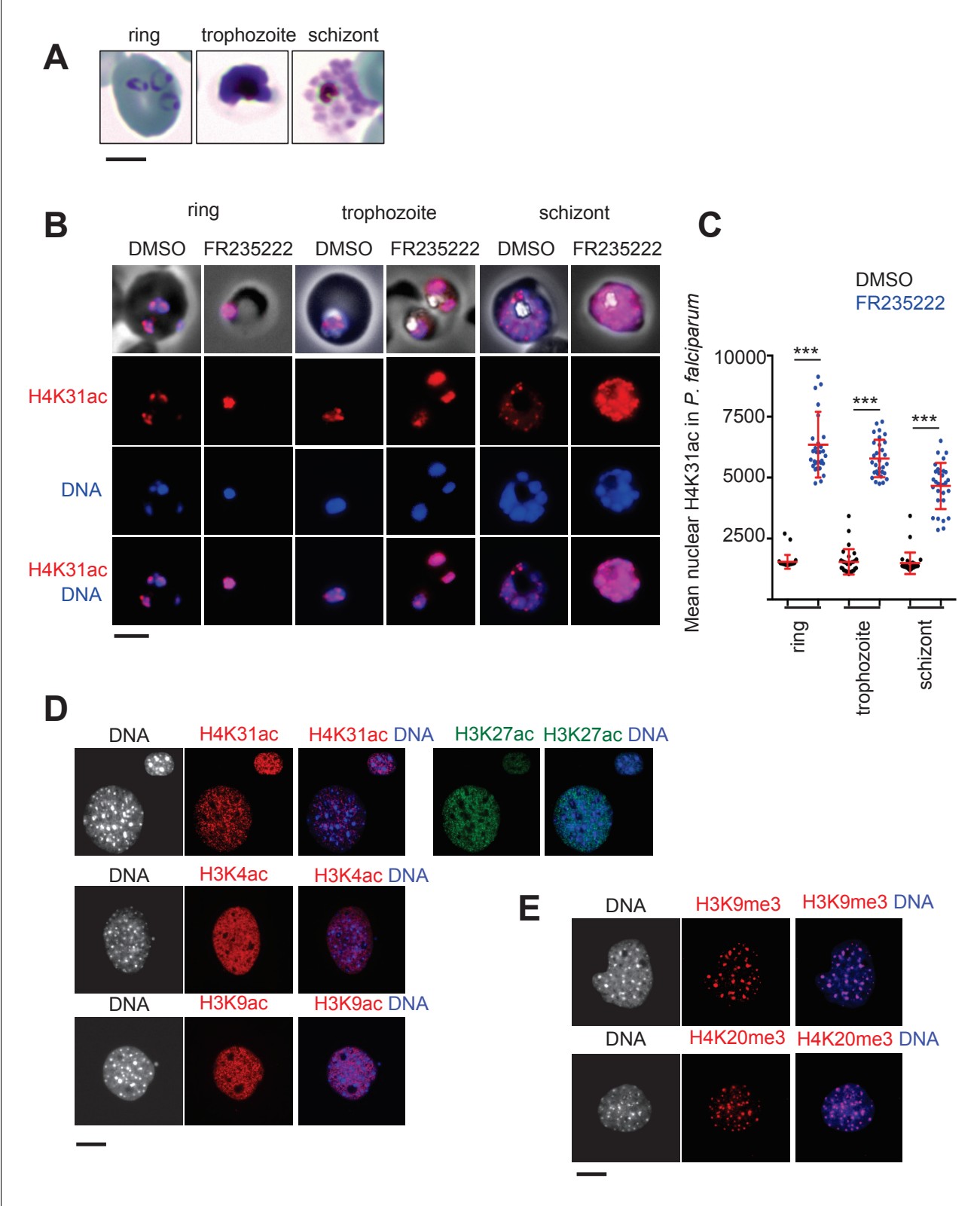

**Figure 3.** Immunofluorescence analysis of histone modifications in *P. falciparum* and mouse embryonic fibroblast (MEF) cells. (a) The blood stages of *P. falciparum* characterized by initial Ring, followed by mature trophozoite and segmented schizont stage. The three developmental stages represent the predominant asexual phase of the malaria parasite. (b) Immunofluorescence analysis of H4K31ac (in red) in asexual stages following 12 hr of treatment with DMSO (vehicle) or FR235222 HDACi. Parasite nuclear DNA was stained with Hoechst (blue). Scale bar, 10 μm. (c) Quantification of the intensity of

*Figure 3 continued on next page*

*Figure 3 continued*

H4K31ac staining in each *P. falciparum* nucleus following FR235222 stimulation of asexual stages. The results are represented as mean ± standard deviations from four independent experiments; the number of nuclei quantified was at least n = 25. Asterisks indicate statistical significance of H4K31ac difference between FR235222-treated sample and the corresponding control (DMSO) in the ring, trophozoite and schizont stages, as determined by an unpaired two-tailed Student's t-test, ***p<0.0001; n.s., not significant. (d) and (e) Immunofluorescence analysis of H4K31ac in MEF. DNA was stained with DAPI (blue); the bright foci mark pericentromeric heterochromatin. The signal for H4K31ac, along with H3K27ac, H3K4ac or H3K9ac, is enriched in euchromatic regions as shown in the merge. The mark is excluded from the DAPI dense foci that are associated with H3K9me3 and H4K20me3. Scale bar, 10 μm. Data are representative of three independent experiments.

DOI: https://doi.org/10.7554/eLife.29391.008

In *P. falciparum*, H4K31me1 displayed a peculiar condensed punctate pattern (*Figure 7c*), similar to that of the H3K9me3 mark (*Lopez-Rubio et al., 2009*) at the nuclear periphery, which is reminiscent of clustering of heterochromatin/subtelomeric regions (*Freitas-Junior et al., 2000*). *P. falciparum* centromeres also clustered prior to and throughout mitosis and cytokinesis, leading to single nuclear location from early trophozoites to mature schizonts (*Hoeijmakers et al., 2012*). Therefore, H4K31me1-containing foci could be associated with subtelomeric and/or centromeric regions. However, the number of foci observed varied, signifying its dynamic changes across parasite developmental stages. The mark was observed in all asexual forms and remained unaffected by treatment with FR235222 (data not shown).

Because H4K31 is also a site of methylation, the transition between H4K31ac and H4K31me1 may represent a novel 'chromatin switch' that contributes to chromatin structure and function in eukaryotic cells. Nevertheless, a different readout is expected from one species to another. In metazoans, H4K31me1 is temporally regulated during the cell cycle and interplay, if any, with H4K31ac should be restricted to mitotic chromosomes. In *P. falciparum*, H4K31me1 formed discrete immuno-fluorescent foci around the nucleus, a pattern quite distinct from that of H4K31ac typified by a diffused signal distributed throughout the parasite nuclei. As H4K31me1 and H4K31ac have distinct nuclear locations and different stoichiometry, H4K31ac being a low-abundance mark, the transition between H4K31me1 and H4K31ac may be not an issue in *P. falciparum* as it can be in *T. gondii* where both modifications are concomitantly distributed throughout euchromatin.

## Nucleosomes with H4K31ac and H4K31me1 are enriched at the promoter and at the core body of active genes, respectively

To further explore whether H4K31ac and H4K31me1 are indeed alternative antagonistic PTMs on the same H4 molecules in *T. gondii*, we examined their genome-wide distributions using chromatin immunoprecipitation coupled with next-generation sequencing (ChIP-seq). We first investigated the relative performance of our homemade antibodies in terms of specificity, sensitivity and the number and distribution of peaks. We observed low variability and a high degree of similarity in read coverage between technical replicates, regardless of the antibodies used for immunoprecipitation (*Figure 8—figure supplement 1*). We next compared the locations of the peaks from each antibody type. Visual display of the chromosomal distribution indicated that H4K31ac and H4K31me1 exhibited distinct patterns of enrichment across the chromosomes and were mutually exclusive genome-wide (e.g., Chr. Ib, *Figure 8a*). Zooming in to detailed gene level revealed that H4K31ac was enriched in distinct peaks at intergenic regions (IGRs) (*Figure 8b*), of which 75% mapped outside the gene body (*Figure 8c*), in line with the euchromatic in situ localization (*Figure 1* and *Figure 3*). The calculated average profile of H4K31ac showed a pattern strikingly similar to that of H3K14ac and H3K4me3, characterized by high signals at 5'UTR/promoter that drop sharply after the translational initiation site (*Figure 8d*). Conversely, H4K31me1 showed a distinct pattern of enrichment, best discernable at large genes, spanning from the ATG to the entire gene body (*Figure 8b*) and absent from IGRs (*Figure 8c*). The H4K31me1 computed average density profile (*Figure 8d*) fully matched with gene prediction, making this mark useful for the explicit detection of unannotated genes (*Figure 8—figure supplement 2*). Remarkably, these data allowed the identification of H4K31me1 as a novel PTM whose spreading was restricted to gene bodies in *T. gondii* (*Figure 8b and d*). However, this mark was not the only one to decorate chromatin in this fashion as a similar pattern was also found for the genome-wide distribution of H3K4me1 in the type I (RH) *T. gondii* strain (Kami Kim, unpublished data available at ToxoDB). We have re-examined the extent and genome-wide

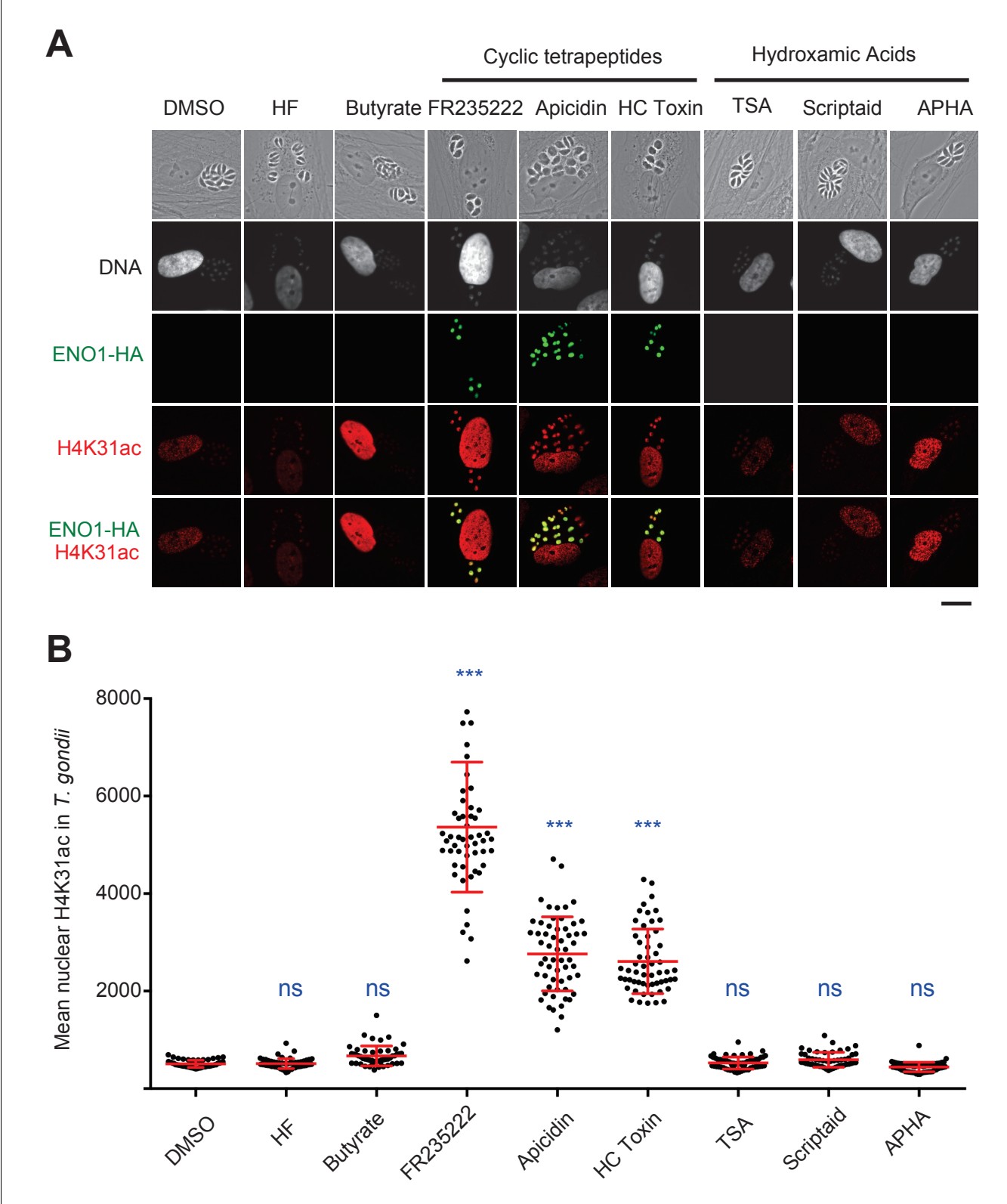

**Figure 4.** Chemical inactivation of TgHDAC3 promotes H4K31ac accumulation in *T. gondii* nuclei. (a) Immunofluorescence analysis of H4K31ac (in red) in HFF cells infected with parasites expressing a HAFlag (HA)-tagged copy of the bradyzoite gene *ENO1* treated for 18 hr with vehicle (DMSO) or individual HDAC inhibitors, including short-chain fatty acids (sodium butyrate), cyclic tetrapeptides and hydroxamic acids. Halofuginone (HF), a non-HDACi anticoccidial compound was used as a relevant control. ENO1 expression was detected by IFA in parasite nuclei (anti-HA, in green). Host-cell

*Figure 4 continued on next page*

*Figure 4 continued*

and parasite nuclei were stained with Hoechst. Scale bar, 20 µm. TSA, Trichostatin A; APHA, aroyl pyrrolyl hydroxamic acid. (**b**) Quantification of the intensity of H4K31ac staining in each parasite nucleus following HDAC inhibitors and vehicle (DMSO) stimulations. Each symbol marks the PTM density of a single parasite nucleus. The results are represented as mean ± s.d. from three independent experiments; the number of nuclei quantified was at least 50. Asterisks indicate statistical for H4K31ac significance between DMSO and each HDACi as determined by a two-way ANOVA with Bonferroni's multiple comparison test, ***$p < 0.0001$ ; n.s., not significant.

DOI: https://doi.org/10.7554/eLife.29391.009

scattering of H3K4me1 using a type II (Pru) genetic background. We observed a high degree of similarity in read coverage between H3K4me1 and H4K31me1 (*Figure 8—figure supplement 3a*) and a similar pattern of enrichment over the gene body (*Figure 8—figure supplement 3b*). However, these PTMs are unevenly distributed across the genome, with H3K4me1 occasionally being weakly enriched relative to H4K31me1 at a few expressed genes (see red dotted square, *Figure 8—figure supplement 3c and d*).

## Interplay between H4K31ac and H4K31me1 predicts distinctive patterns of gene expression in *T. gondii*

A closer view of H4K31ac and H4K31me1 chromosomal binding revealed that, at some loci (e.g., *SRS* family genes [*Figure 9a*] and *GRA1* [*Figure 9b*]), the latter was absent whereas the former was enriched at the 5'UTR/promoter and unexpectedly spread over a much larger area, overlapping the gene body (see *GRA1* and *MAG1* examples, *Figure 9c*). The restricted H4K31ac enrichment in the vicinity of *GRA1 or SRS* genes contrasts with the H4K31ac and H4K31me1 location on their neighboring genes (*TGME49_233490* or *TGME49_270230*) and this discrepancy may be explained by the higher level of *SRS* or *GRA1* gene expression (*Figure 9a and b*). We therefore interrogated whether the enrichment patterns of modified H4K31 could specify levels of gene expression in *T. gondii*. We first conducted a global transcriptome analysis by RNA-Seq of tachyzoites during growth phase in murine bone marrow-derived macrophages (BMDMs). Cluster analysis revealed varying levels of gene expression with cluster Q1 displaying the highest level, clusters Q2 and Q3 defining intermediate levels, and cluster Q4 displaying the lowest level (*Figure 9d* and *Figure 9—source data 1*). High mRNA level (in Q1, which includes the *GRA1, MAG1* and *SRS* genes) was associated with a high level of H4K31ac upstream of the ATG together with an enrichment along the gene body, which coincided with the expected lack of H4K31me1 (*Figure 9e and f*). In highly expressed relatively long or intron-containing genes (e.g., *MAG1*), H4K31ac spread but did not extend over the entire gene body as observed for *GRA1* (*Figure 9c*), indicating a limited spreading of H4K31 acetylation around the translational initiation site. Strikingly, moderate mRNA levels (in Q2 and Q3, which include *TGME49_233490* or *TGME49_270230*) related to a relatively high level of H4K31me1 in the gene body and a restricted mapping of H4K31ac at the promoter, thereby arguing for an inverse correlation between the yield of expression and the level of H4K31 methylation (*Figure 9e and f*).

Finally, transcriptionally repressed genes clustered in Q4 showed no significant enrichment of either H4K31ac or H4K31me1 (*Figure 9e and f*) but were enriched in the repressive mark H3K9me3 (*Figure 9—figure supplement 1e*). In addition, we discovered that genes clustered in Q4 were transcriptionally heterogeneous and that the repressive signature H3K9me3 was primarily present in the vicinity of genes typified by low (if not undetectable) RNA levels in tachyzoites and referenced as stage-specific within the *T. gondii* life cycle. As such, the *ENO1* and *BAG1* genes, which are well known to be expressed distinctively in the bradyzoite stage during the chronic phase of infection (*Pittman et al., 2014*), were highly enriched in the repressive mark H3K9me3 but they were also unexpectedly enriched in the gene activation hallmark H3K14ac (*Figure 10a* and *Figure 10—figure supplement 1a*). Coccidian-specific surface gene *SRS* families, whose expression was restricted to the bradyzoite stage (*Figure 10—figure supplement 1b*), also displayed marks of both active and silent chromatin (*Figure 10—figure supplement 1c and d*). This dual histone PTM was also detected on nucleosomes surrounding genes that have been recognized as being expressed exclusively during sexual stages in the definitive feline host (*Hehl et al., 2015*). The co-enrichment of H3K9me3 and H3K14ac was somehow restricted to the transcriptional units that become active in the sexual stages (shown in red, *Figure 10b*). However, H3K14ac was seen to spread over some tachyzoite-expressed neighboring genes (shown in blue, *Figure 10b*) and probably contributes to

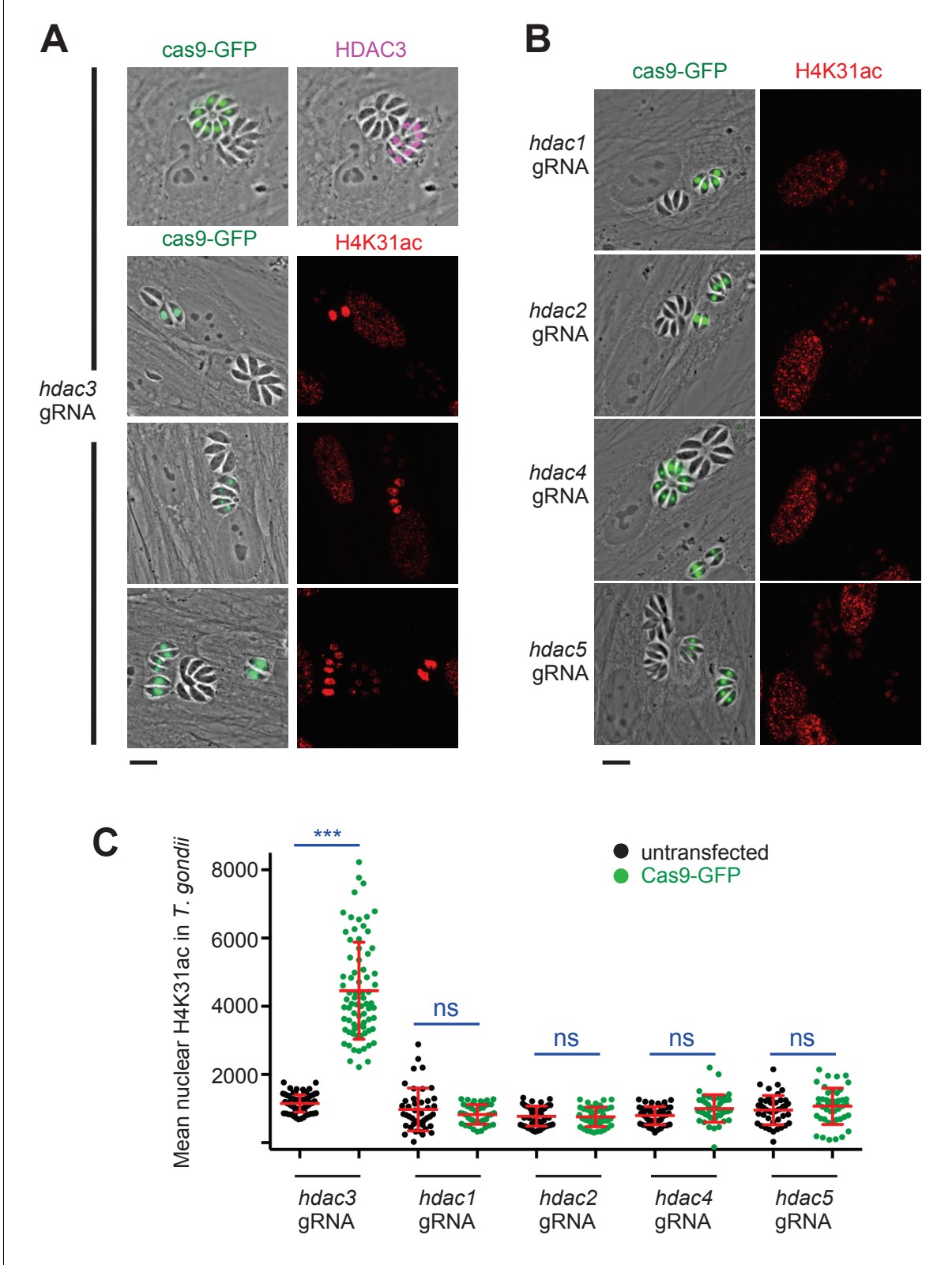

**Figure 5.** Genetic inactivation of TgHDAC3 promotes H4K31ac accumulation in *T. gondii* nuclei. (a) Representative micrographs showing intracellular tachyzoites in which the *TgHDAC3* gene was disrupted by transient transfection of CRISPR/Cas9. The efficiency of TgHDAC3 disruption in Cas9-expressing parasites was monitored by anti-TgHDAC3 staining (in pink) and cas9-GFP expression (in green). The levels of H4K31ac (in red) were monitored in *TgHDAC3*-disrupted parasites (green fluorescent protein [GFP]-positive) and compared to untransfected parasites (GFP-negative). Scale

*Figure 5 continued on next page*

*Figure 5 continued*

bar, 10 μm. (c) Levels of H4K31ac (in red) were monitored in *TgHDAC*-knockout parasites. Scale bar, 10 μm. Data are representative of four independent experiments. (b) Quantification of the intensity of the H4K31ac nuclear staining of *T. gondii* expressing gRNA directed against TgHDAC as indicated (GFP-positive cells) or left untransfected. Each symbol marks the H4K31ac density of a single parasite nucleus. The results are represented as mean ± s.d. from four independent experiments; the number of nuclei quantified was at least 60. Asterisks indicate statistical significance for H4K31ac difference between cas9-GFP-positive and untransfected parasites, as determined by an unpaired two-tailed Student's t-test, ***p<0.0001 ; n.s., not significant.

DOI: https://doi.org/10.7554/eLife.29391.010

their activation. Genes carrying this chromatin signature fall into two transcriptomic clusters within the enteroepithelial stages (*Hehl et al., 2015*): (1) a cluster corresponding to the merozoite genes, i.e. those expressed in parasites harvested from cat at day 3 (*Figure 10—figure supplement 2a and b*) which include *GRA11b* (*Ramakrishnan et al., 2017*); and (2) a cluster of genes typical of the sexual stages, among which *AP2X-10* (*Hong et al., 2017*) and *SUB6* are representative examples (*Figure 10—figure supplement 2c and d*). Merozoites, which only infect the feline enterocytes, were shown to express a large repertoire of 52 SRS proteins specifically (*Hehl et al., 2015*). Using the 111 members of the SRS superfamily of proteins annotated in ToxoDB, we identified 66 *SRS* genes displaying H3K9me3/H3K14ac enrichment, including the aforementioned 52 genes that are specific to merozoites (*Figure 10—figure supplement 3*) along with 8 bradyzoite *SRS* genes (*Figure 10—figure supplement 1b*). By contrast, the 14 *SRS* genes exclusively expressed in tachyzoites (e.g. the *SAG1* cluster) lack the dual PTM (*Figure 9a*).

The co-enrichment of H3K14ac and H3K9me3 is somewhat paradoxical. Although the reasons for this pattern remains unclear, it seems to bookmark genes that are repressed temporally, which await parasite-stage differentiation for stage-specific expression. Thus, H3K14ac and H3K9me3 could form the so-called bivalent chromatin domain capable of silencing developmental genes while keeping them poised for rapid activation upon cell differentiation (*Voigt et al., 2013*). The H3K14ac along with the repressive mark H3K27me3 have been shown to be enriched in a similar manner at a subset of inactive promoters in mouse embryonic stem cells (*Karmodiya et al., 2012*). Bivalent domains have gathered wide attention because they might contribute to the precise unfolding of gene expression programs during cell differentiation. Apparently, these bivalent marks are restricted to inactive stage-specific promoters, which differ from both pericentromeric (*Figure 10c* and *Figure 10—figure supplement 4*) and telomeric (*Figure 10d*) heterochromatin which are decorated by H3K9me3 but missing H3K14ac enrichment. It was previously reported that H3K9me3 typifies centrometric heterochromatin in *T. gondii* (*Brooks et al., 2011*) but this study conflicts with our data in reporting enrichment of the mark to 'poised' stage-specific genes. This discrepancy could be explained by the genetic background of the parasite strain, as Brooks and colleagues infected human cells with a type I (RH) strain that lost the ability to develop into mature cysts whereas we used infections with a type II strain (Pru), which is more relevant as it does readily develop tissue cysts and latent infections in animals.

Taken together, our data highlight unique chromatin signatures that are associated with transcription rate in *T. gondii*. Genes clustered in Q1 are primarily defined by low H4K31me1 enrichment and enhanced acetylation at both promoter and the 5' proximal gene body, whereas those from Q2 and Q3 are markedly typified by the presence of H4K31 methylation in the gene body and an acetylation mark restricted to the promoter. In this context, H4K31ac would be predicted to disrupt histone–DNA interaction, thereby affecting nucleosome stability while promoting RNA polymerase progression across transcribed units. Conversely, H4K31me1 could act as a transcription-linked repressive mark that may hypothetically slow the progress of the RNA polymerase on active genes, proabbly by modulating the transcription-dependent histone turnover, but this still needs to be established. Nevertheless, the mark does not elicit its predicted repressive effect on constitutively repressed genes. Otherwise, we identified a subset of genes that were typified by their exclusive expression in chronic and sexual stages that are displaying typical bivalent chromatin domains characterized by H3K9me3 and H3K14ac enrichments in acute-phase tachyzoites.

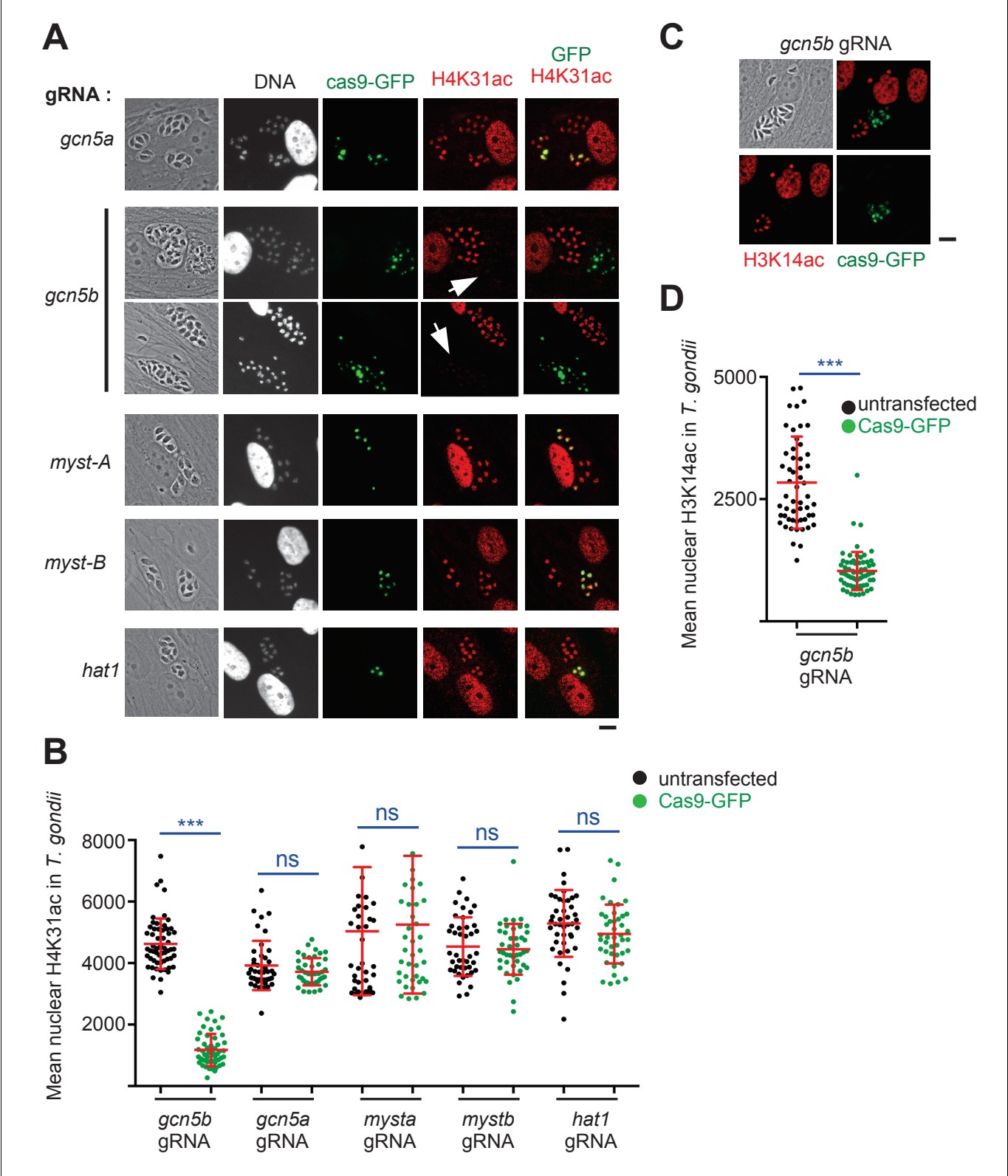

**Figure 6.** TgGCN5b acetylates H4K31 in *T. gondii*. (a) Levels of H4K31ac (in red) were monitored in both host-cell and parasite nuclei following CRISPR/Cas9-mediated disruption of individual *T. gondii* HAT enzymes. Transfected vacuoles in which H4K31 acetylation was impaired are indicated by a white arrows. Scale bar, 10 μm. (b) Quantification of the intensity of the H4K31ac nuclear staining of *T. gondii* expressing the gRNAs directed against TgHAT (GFP-positive cells) or left untransfected. Each symbol marks the H4K31ac density of a single parasite nucleus. The results are represented as
*Figure 6 continued on next page*

*Figure 6 continued*
mean ± s.d. from three independent experiments; the number of nuclei quantified was at least 40. Asterisks indicate statistical significance of H4K31ac signal differences between cas9-GFP-positive and untransfected parasites as determined by an unpaired two-tailed Student's t-test, ***p<0.0001; n.s., not significant. (c) Levels of H3K14 acetylation (in red) were monitored in *TgGCN5b*-knockout parasites. Scale bar, 10 µm. (d) Quantification of the intensity of the H3K14ac nuclear staining of *T. gondii* expressing the gRNAs directed against *gcn5b* (GFP-positive cells) or left untransfected. Data are representative of three independent experiments. ***p<0.0001 (unpaired two-tailed Student's t-test).
DOI: https://doi.org/10.7554/eLife.29391.011

## H4K31me1 enrichment, a blueprint for unannotated genes and uncharacterized long non-coding RNAs

As mentioned previously, H4K31me was mainly detected throughout the body of active genes with translation start and stop codons as boundaries, and its enrichment was inversely correlated to the yield of mRNA. These features should allow this mark to predict unannotated genes explicitly, even when the low level of expression impedes detection by RNA profiling (*Figure 8—figure supplement 2*).

Although H4K31me1 rarely covered IGRs (*Figure 8c*), the mark was occasionally found to be enriched in chromosomal regions that were devoid of any predicted protein-coding genes (*Figure 10d* and *Figure 10—figure supplement 5*). This enrichment correlated with extensive transcription of large RNA transcripts, ranging from 20 to 70 kB, reminiscent of long noncoding RNAs (lncRNAs) in other eukaryotic cells (*Azzalin and Lingner, 2015*). These *T. gondii* lncRNAs are stand-alone transcription units with a proper chromatin signature, i.e., H4K31ac and H3K4me3 at the promoter and H4K31me1 along the transcribed length (*Figure 10d* and *Figure 10—figure supplement 5*). Considering their distribution at both telomere-adjacent regions (*Figure 10d*) and chromosome arms (*Figure 10—figure supplement 5*), these lncRNAs may work in cis near the site of their production (e.g., they may function in telomere homeostasis) or act in trans to alter chromatin shape and gene expression at distant loci, as reported in other model organisms (*Azzalin and Lingner, 2015*).

## Distribution of H4K31 modifications across the *P. falciparum* genome reveals H4K31me1 as a novel pericentromeric PTM

The *P. falciparum* genome is primarily maintained in a decondensed euchromatic state with perinuclear heterochromatin islands. These heterochromatin-based gene silencing regions are used for the regulation of monoallelic expression of clonally variant genes (e.g. *var* and *rifin*) and are enriched in H3K9me3, which binds HP1 (*Voss et al., 2014*). We observed an apparent non-overlapping staining for acetylated and methylated H4K31 and more specifically a discrete focal distribution of H4K31me1 at the nuclear periphery (*Figure 3b* and *Figure 7c*). To get a comprehensive view of the genomic distribution of these PTMs across the *P. falciparum* genome, we also performed ChIP-seq analyses during the IDC. As for *T. gondii*, we observed low variability and high similarity in read coverage between technical replicates for all the antibodies used (*Figure 11—figure supplement 1*).

We next compared the peak locations for each antibody type. Like the euchromatic mark H3K4me3, H4K31ac displayed a rather even distribution throughout the genome (*Figure 11a* and enhanced view at *Figure 11—figure supplement 2*). As for *T. gondii*, the distribution of H4K31ac in *P. falciparum* matched with the gene annotation, i.e., high acetylation rates at promoters and low rates at the gene body of active genes (e.g., *GAPDH*, *Figure 11b*). Consistent with this, the presence of H4K31ac and that of the repressive mark H3K9me3 were inversely correlated (*Figure 11a*). However, whereas H4K31ac displayed a relatively narrow enrichment restricted to transcribed promoters, H3K4me3 was enriched in a large fraction of the genome (*Figure 11c*) as already described (*Salcedo-Amaya et al., 2009*).

Interestingly, the methylation of H3K9 and the HP1-binding properties that have emerged as hallmarks of pericentromeric heterochromatin in model systems including *T. gondii* (*Brooks et al., 2011*; *Gissot et al., 2012*) have not been detected in *P. falciparum*, leading to the view that the parasite may lack pericentric heterochromatin (*Flueck et al., 2009*; *Lopez-Rubio et al., 2009*; *Salcedo-Amaya et al., 2009*). Although our ChIP-seq analysis confirmed the absence of pericentric enrichment of both H3K9me3 and HP1, it clearly highlighted a remarkable enrichment of H4K31me1 at

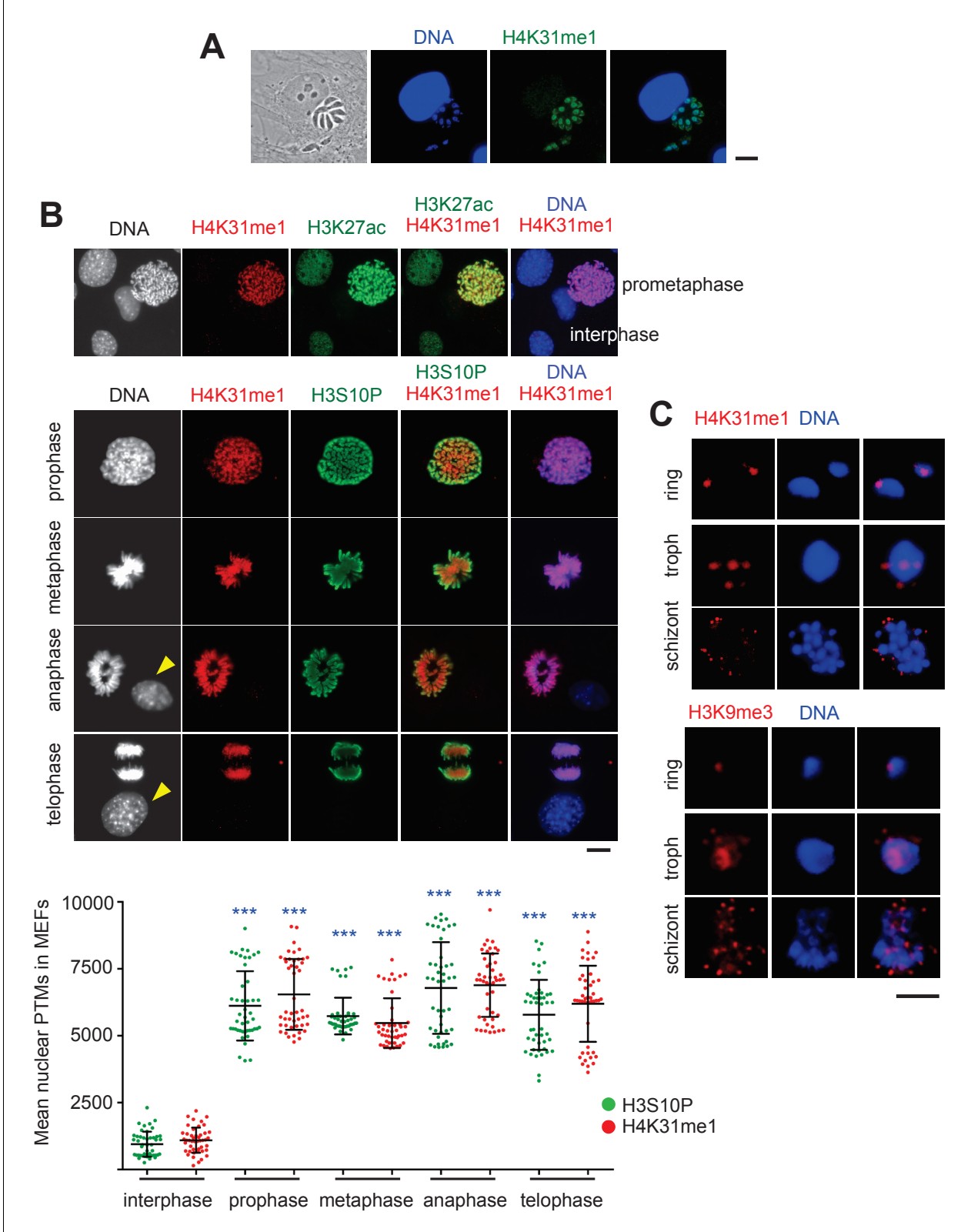

**Figure 7.** Distribution of H4K31me1 in host-cell and parasite nuclei. (a) Immunofluorescence analysis of H4K31me1 (in green) in intracellular parasite nuclei. DNA was stained with Hoechst (blue). Scale bar, 10 μm. (b) The localization of H4K31me1 (in red), H3K27ac (in green) and H3S10P (in green) were tested by immunofluorescence in prophase, metaphase, anaphase and telophase MEFs. DNA was stained with Hoechst (blue). Scale bar, 10 μm. Quantification of H4K31me1 and H3S10P staining for each MEF analysed in its respective phase of the cell cycle. In total, 45 cells were scored for each

*Figure 7 continued on next page*

*Figure 7 continued*

phase. Each dot represents a single cell nucleus. The results are represented as mean ± s.d. from three independent experiments. Asterisks indicate statistical significance of the difference in H4K31me1 or H3S10P signal between interphase (background control) and each phase of the cell cycle as determined by a two-way ANOVA with Bonferroni's multiple comparison test, ***p<0.0001; n.s., not significant. (c) Immunofluorescence analysis of H4K31me1 (in red) or H3K9me3 in asexual stages of Pf-3D7. Scale bar, 5 μm. Data are representative of three independent experiments.
DOI: https://doi.org/10.7554/eLife.29391.012

pericentromeric regions that flank the cenH3-enriched centromeres (*Figure 11d*, *Figure 11—figure supplement 3* and *Figure 12a*). It is therefore possible that H4K31me1 constrains PfCENH3 to the centromeres in *P. falciparum*, thus replacing the H3K9me3–HP1 functions described in most species. In addition to the pericentromeric localization, H4K31me1 was also enriched at a few subtelomeric regions and more specifically at the telomere-associated repetitive element (TARE, *Figure 12b*) repeat blocks shown to encode lncRNAs (*Figure 12c*) (*Sierra-Miranda et al., 2012*). The presence of the mark at pericentric chromatin combined with its absence at transcriptionally permissive loci (e.g. *GAPDH*, *Figure 11b*) suggest H4K31me1 as a novel hallmark of heterochromatin in *P. falciparum*, but this mark is not similar to H3K9me3–HP1 in subtelomeric regions and at *var* genes.

## Discussion

This study provides an in depth understanding of the interaction between the core histone H4 and the template DNA by functionally characterizing novel modifications of H4K31, a residue exposed on the outer surface of the nucleosome in close proximity to the DNA entry-exit point. Proteome-wide mapping of acetylome/methylome as well as nucleosome protein content analyses allowed the identification of H4K31 as a site for both acetylation and methylation across a wide range of species, including those from the apicomplexan parasitic phylum. The K31 residue lies at the N-terminus of the histone H4 α1 helix, and its positively charged side chain forms water-mediated interactions with the DNA phosphate backbone (*Figure 1c*). Its acetylation was predicted to trigger substantial conformational changes in the nucleosome by shifting the side chain of lysine from the unacetylated to the acetylated state, causing the loss of the water-mediated interactions that K31 establishes with DNA and the residue R35 (*Figure 1c*). However, this prediction was not validated as X-ray crystallography did not indicate large structural changes to nucleosomes when glutamine was substituted for lysine to mimic the acetylated state (H4Q31, *Figure 1c*) (*Iwasaki et al., 2011*). Alternatively, H4K31ac may increase DNA unwrapping at the entry-exit point of the nucleosome, thus giving access to the ATP-dependent chromatin remodelers that act on nucleosome disassembly and turnover as proposed by *Chatterjee et al. (2015)*. The latter assumption would fit with the 'regulated nucleosome mobility' model (*Cosgrove et al., 2004*), which predicts that outer surface PTMs (e.g., H3K36ac [*Williams et al., 2008*]) regulate the equilibrium between mobile and relatively stationary nucleosomes by altering histone–DNA molecular interplay.

In both *T. gondii* and *P. falciparum*, genome-wide studies pinpointed a local enrichment of H4K31ac at active gene promoters, in line with the cooperative contribution of acetylation and other PTMs in shaping a transcriptionally permissive chromatin state. H4K31ac relieves nucleosomal repression thus facilitating the access of the transcriptional machinery to the DNA template, whereas H4K31 monomethylation probably locks the nucleosome in a repressed conformation that maintains chromatin in a closed or semi-closed state also called poised-state, while it also prevents GCN5-related HAT from catalyzing the acetylation of the residue. Interestingly, apart from its predicted effects on the nucleosome mobility and chromatin state, we found that, in *T. gondii*,H4K31ac also prevents methylation at the body of highly expressed genes, thereby ensuring maximal efficacy of the RNA polymerase progression and activity. Indeed, we found H4K31me1 enrichment only in the transcribed coding sequence of a subset of genes typically associated with limited activity of the RNA polymerase II. In a model where the nucleosome disassembles in front of transcribing RNA polymerase II to allow its physical progression across transcribed units, it is plausible that, by stabilizing the wrapping of DNA around the histone octamer, H4K31me1 slows down RNA Pol II processing along the fiber and thus reduces the level of transcription.

Aside from specific patterns of PTMs, histone chaperones significantly contribute to controlling how the RNA polymerase II engages the nucleosome in and around a promoter and during the

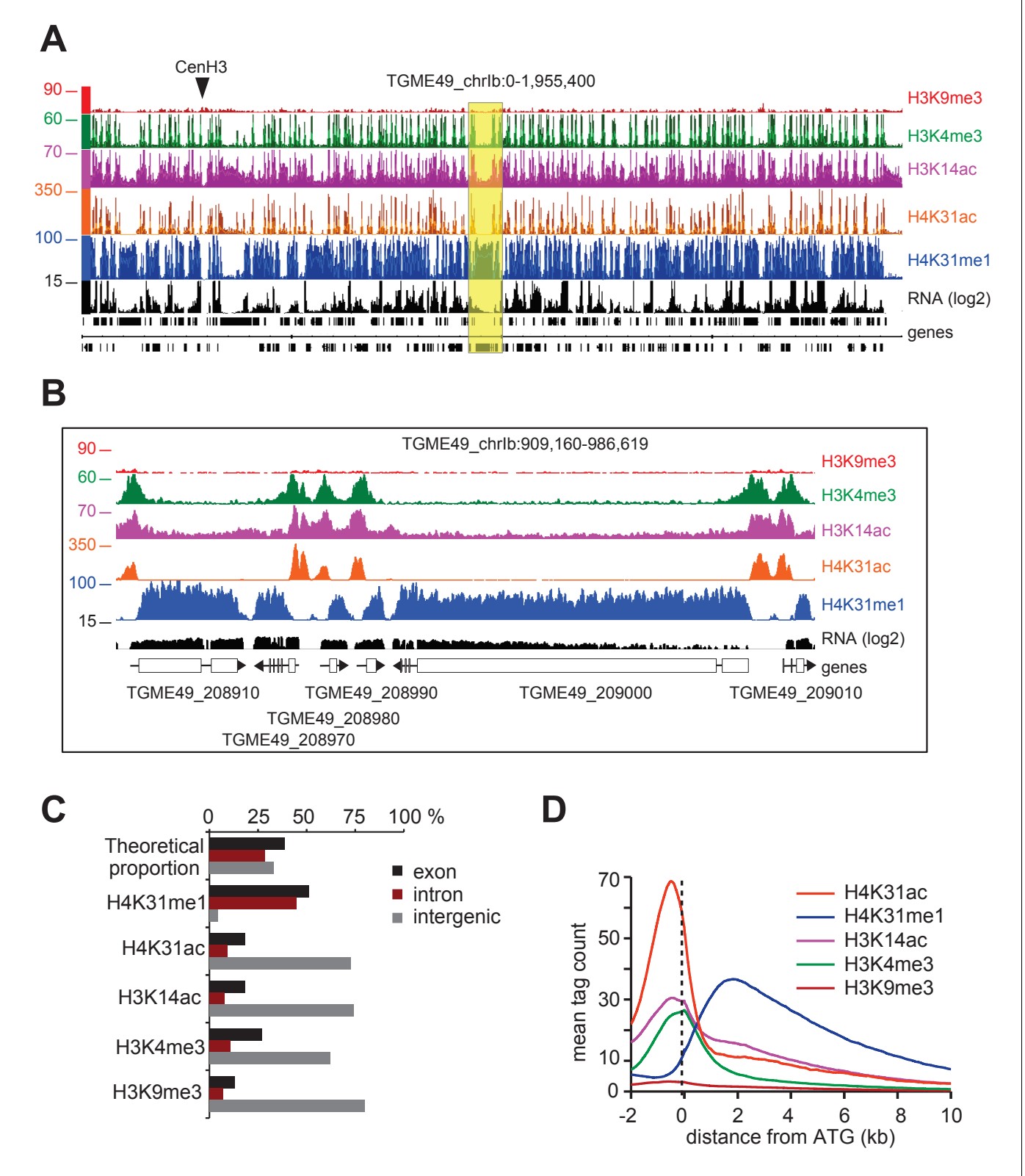

**Figure 8.** Genome-wide analysis of H4K31ac and H4K31me1 chromatin occupancy in *T. gondii*. (**a**) A genome browser (IGB) snapshot showing normalized reads for different histone marks across *T. gondii* chromosome 1b reveals a peak-like distribution of H4K31ac and H4K31me1 ChIP-seq enrichments. The y-axis depicts read density. Genes are depicted above the profiles in black. (**b**) A zoomed-in view of a Chr. Ib region (yellow box in **a**) showing the distribution of the aforementioned PTMs. (**c**) Distribution of PTM-occupied regions relative to the *T. gondii* reference genome annotation. *Figure 8 continued on next page*

*Figure 8 continued*

(d) Correlation of the enrichment of H4K31 modifications with that of other marks. The average signal profiles of each histone modification was plotted over a −2 kb to +10 kb region with respect to *T. gondii* genes' ATG. The y-axis shows the average tag count of the enrichment. The vertical dashed line indicates the position of the ATG.

DOI: https://doi.org/10.7554/eLife.29391.013

The following figure supplements are available for figure 8:

**Figure supplement 1.** ChIP-seq enrichments in biological replicates are highly correlated.

DOI: https://doi.org/10.7554/eLife.29391.014

**Figure supplement 2.** H4K31me1 explicitly predicts unannotated genes.

DOI: https://doi.org/10.7554/eLife.29391.015

**Figure supplement 3.** H4K31me1 and H3K4me1 are both enriched in gene bodies in *T. gondii*.

DOI: https://doi.org/10.7554/eLife.29391.016

elongation step. For instance, the FACT (Facilitates Chromatin Transactions) histone chaperone has been shown to assist first in the removal of nucleosomes ahead of the transcribing RNA Pol II and next in their reassembly after polymerase passage. Although we provide evidence that H4K31 modifications may contribute to gene regulation at least in *T. gondii*, studies in other eukaryotes have underlined H4K31's instrumental role in the recruitment or mobilization of histone chaperone at transcribed genes. In budding yeast, H4K31 along with two proximal residues on the side of the nucleosome (i.e., H4R36 and H3L61) promotes the recruitment of the yFACT subunit Spt16 across transcribed genes, as assessed by the typical change in Spt16 distribution in which occupancy shifts toward the 3' ends of transcribed genes in the H4K31E yeast mutant (*Nguyen et al., 2013*).

The versatility of H4K31 goes beyond even these modifications as ubiquitylation of H4K31 has been reported as an additional regulatory PTM for transcription elongation in human cells (*Kim et al., 2013*). Indeed, it has been shown that the histone H1.2 subtype, while localized at target genes, regulates the elongating RNA Polymerase II in an interaction that is typified by phosphorylation of Ser2 on its carboxy-terminal domain (CTD). Indeed, it was shown that upon interaction with the Ser2-phosphorylated CTD of the active RNA Pol II, the histone H1.2 subtype becomes able to recruit the Cul4A E3 ubiquitin ligase and PAF1 elongation complexes. In turn, those proteins stimulate H4K31 ubiquitylation that positively influences the accumulation of the H3K4me3/H3K79me2 signature, thereby leading to a more productive elongation phase of transcription. Importantly, blocking H4K31 ubiquitylation by K31R mutation markedly reduces H3K4 and H3K79 methylation and consequently impairs gene transcription (*Kim et al., 2013*).

In order to test the functional significance of H4K31 modifications in vivo, we tried to substitute H4K31 in the *T. gondii* genome with alanine or glutamine to mimic acetyl lysine or with arginine to mimic nonacetylated lysine but were unsuccessful in achieving these substitutions (data not shown). Engineered budding yeast with those substitutions did not significantly affect cell viability but led to an unexpected increase of telomeric and ribosomal DNA silencing (*Hyland et al., 2005*), both of which argue for the mutations' driving a non-permissive chromatin state. This finding does not fit with our working model in whichH4K31Q should promote open chromatin. It is plausible, however, that the substitutions did not faithfully mimic the effects of the modifications in these instances. In sharp contrast with the aforementioned substitutions, glutamic acid (E) that mimics succinylated lysine was shown to compromise the growth in budding yeast severely (*Xie et al., 2012*), maybe as a consequence of the alteration in the distribution of Spt16 across yeast genes (*Nguyen et al., 2013*). Succinylation on H4K31 has also been detected by mass spectrometry in *T. gondii* (*Li et al., 2014*; *Nardelli et al., 2013*). This modification could drastically impact intranucleosomal structure and induce 'abnormal' histone–DNA interactions (*Figure 1c*).

In this context, H4K31 methylation would counteract the activating effect of H4K31 acetylation and succinylation, by preventing the nucleosome from adopting an open conformation that permits gene expression. The analysis in *P. falciparum* revealed remarkable features of H4K31me1 by stressing a much more pronounced repressive character because the modification was exclusively restricted to non-permissive silenced chromosomal zones. Originally, *P. falciparum* heterochromatin — in which clusters of genes are maintained in a silent state — was singularly defined by increased nucleosomal occupancy, histone deacetylation, H3K9me3 and the binding of PfHP1 (*Scherf et al., 2008*). Most of the genome can be characterized as euchromatin, but those silenced

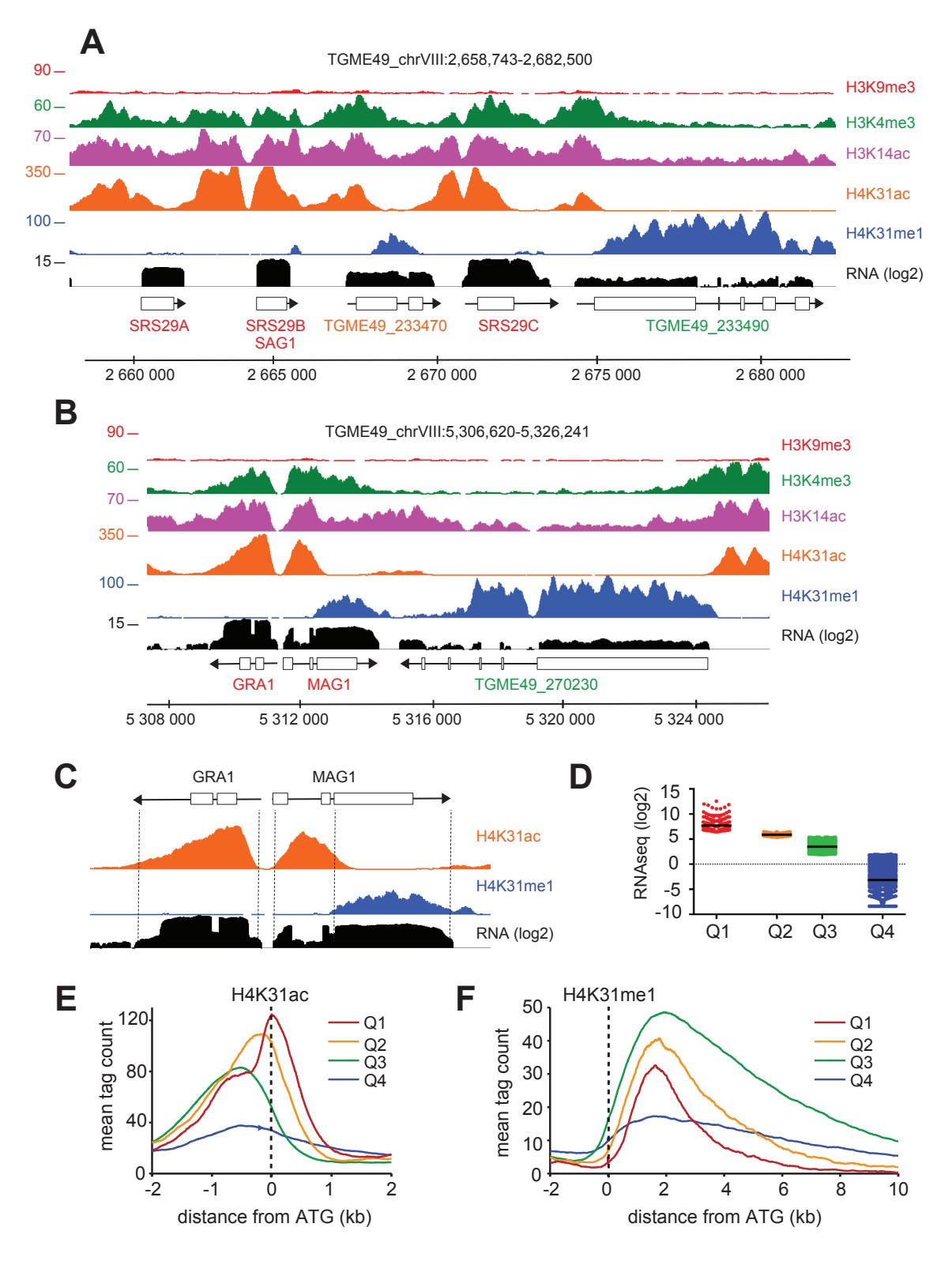

**Figure 9.** The enrichment of H4K31ac and H4K31me1 at transcribed genes correlates with gene expression levels in *T. gondii*. (a, b) IGB screenshots of *T. gondii* Chr. VIII genomic regions showing reads for various histone marks as well as RNA-seq data (in black). (c) A zoomed-in view of the *T. gondii GRA1-MAG1* locus. The y-axis depicts read density. (d) Boxplot showing the normalized expression distribution of *T. gondii* genes in the tachyzoite stage subdivided into four ranges of expression (clusters Q1 to Q4). Genome-wide H4K31ac (e) and H4K31me1 (f) occupancy profiles at peri-ATG

*Figure 9 continued on next page*

*Figure 9 continued*

regions are plotted for the gene groups ranked by their mRNA levels. The y-axis shows the average tag count of the enrichment. The vertical dashed line indicates the position of the ATG.

DOI: https://doi.org/10.7554/eLife.29391.017

The following source data and figure supplement are available for figure 9:

**Source data 1.** Table corresponding to *Figure 9D* containing the normalized expression distribution of *T. gondii* genes in the tachyzoite stage subdivided into four ranges of expression (clusters Q1 to Q4).

DOI: https://doi.org/10.7554/eLife.29391.019

**Figure supplement 1.** PTM distribution and gene expression in *T. gondii*.

DOI: https://doi.org/10.7554/eLife.29391.018

regions were organized towards the periphery of the nucleus and contain, among others, the *var*, rif and stevor families that cluster together, proximal to each telomere. The repression of the *var* genes, for instance, involves the trimethylation of H3K9 and its spreading to the next-door nucleosome by the action of HP1 (*Scherf et al., 2008*). H4K31me1 enrichment was detected, although unevenly and at low rates, in the vicinity of few *var* (5 out of 64) and *rifin* genes (*Figure 12c*). However, the modification does not spread while its enrichment fades quickly and probably remains limited to the site of heterochromatin initiation where both H3K9me3 and HP1 levels culminate (*Figure 12c*). The lack of spread of H4K31me1 along a series of nucleosomes may be explained by the absence of a competent protein reader that specifically recognizes the PTM and recruits the H4K31me1-catalyzing methyltransferase. To date, no H4K31me1-reading protein has been identified, although the PTM is not buried and hence is accessible for the binding of regulatory factors. In fact, the bromodomain of BRD4 is able to recognize the acetylated isoform of H4K31 (*Filippakopoulos et al., 2012*).

Although H4K31me1 occupancy is generally limited across the *P. falciparum* genome, the modification is by far the most promiscuous PTM found at pericentromeric zones of all chromosomes (*Figure 11d* and *Figure 12—figure supplements 1* and *2*). As such, both H4K31me1 (*Figure 7c*) and centromeres (*Hoeijmakers et al., 2012*) were found to be clustered towards the nuclear periphery. *P. falciparum* centromeres were originally described as displaying a unique epigenetic status typified by the noteworthy absence of the canonical pericentromeric PTM H3K9me3 (*Hoeijmakers et al., 2012*), which is present in all species including *T. gondii* (*Figure 10c* and *Figure 10—figure supplement 4*) (*Brooks et al., 2011*). Clearly this study has emphasized an unusual role of H4K31me1 in pericentromeric heterochromatin in *P. falciparum* and has provided new insights on the mechanism of transcriptional regulation in *T. gondii*.

In metazoans, H4K31me1 was shown to decorate the mitotic chromosome arms (*Figure 7b*). In this regard, H4K31me1 is a novel mitotic marker that targets newly synthesized H4 that may be involved in the regulation of chromosomal condensation and segregation during mitosis. H4K31 is structurally very close to H3K56 (*Figure 1c*), the acetylation of which is reported to increase the binding affinity of H3 toward histone chaperones, thereby promoting nucleosome assembly during the S phase of the cell cycle (*Li et al., 2008*). Collectively, our results argue for a similar role for H4K31me1 in chromatin assembly during DNA replication in metazoans. However, the picture appears more complex than this as H4K31 methylation, unlike H3K56ac, is predicted to prevent histone exchange, thereby slowing histone turnover rate behind the replication forks, which overall contributes to the stabilization of newly incorporated nucleosomes in chromatin.

In conclusion, we demonstrate that H4K31 acetylation and methylation are associated with very distinct nuclear functions in *T. gondii* and *P. falciparum*. Moreover, we demonstrate the evolution of distinct epigenetic strategies to organize chromosome regions that are essential for cell division and gene expression in apicomplexan parasites.

## Materials and methods

### Parasites and host cells culture

HFF primary cells (*Bougdour et al., 2009*) were cultured in Dulbecco's Modified Eagle Medium (DMEM) (Thermo Fischer Scientific, France) supplemented with 10% heat-inactivated Fetal Bovine Serum (FBS) (Invitrogen), 10 mM (4-(2-hydroxyethyl)−1-piperazine ethanesulphonic acid) (HEPES)

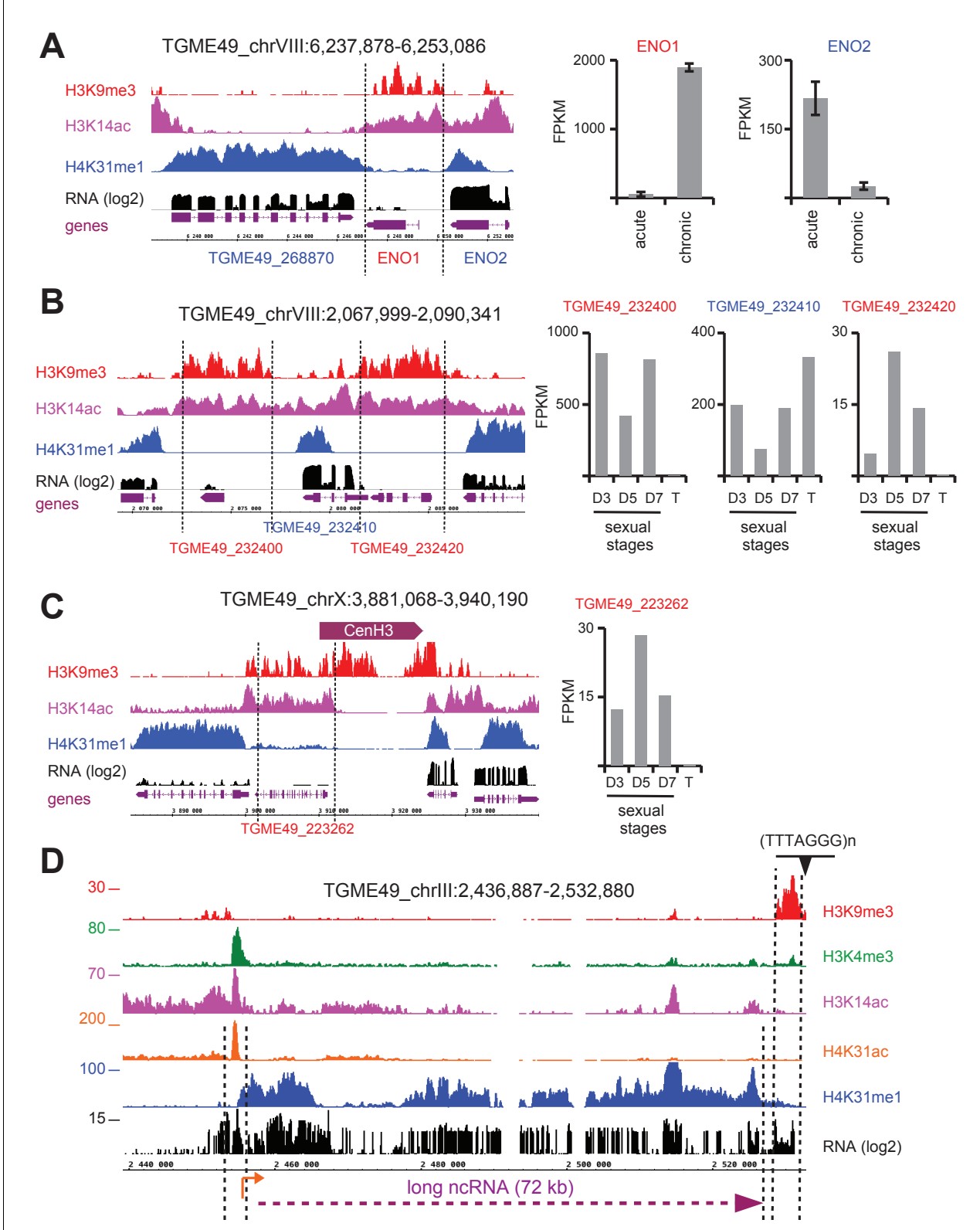

**Figure 10.** H4K31me1 marks long non-coding RNA but not pericentromeric heterochromatin in *T. gondii*. (a) Left panel : IGB screenshot of *T. gondii* *ENO1* and *ENO2* loci on Chr. VIII , showing reads for various histone marks as well as RNA-seq data (in black) and predicted genes (in magenta). The y-axis depicts read density. Right panel: bar graphs showing the expression (fragments per kilobase of transcript per million mapped reads [FPKM] values) of *ENO1* and *ENO2* genes during acute (tachyzoite) or chronic (bradyzoite) infection in mice (data source: ToxoDB [*Pittman et al., 2014*]). (b) *Figure 10 continued on next page*

*Figure 10 continued*

Left panel: IGB screenshot of two sexual stages genes (in red) surrounding a house-keeping gene (in blue). Right panel: bar graphs showing expression (FPKM values) of the genes in cultured tachyzoites (T) and parasites harvested from cat at day 3 (D3; merozoites), day 5 (D5; merozoites and sexual stages) and day 7 (D7; sexual stages and oocysts) (data source : ToxoDB [*Hehl et al., 2015*]). (c) A zoomed-in view of the *T. gondii* Chr. X peri-centromeric region and neighboring genes including *TGME49_223262,* whose expression is restricted to sexual stages as seen in the bar graph. (d) IGB screenshot of the *T. gondii* Chr. III genomic region showing reads for various histone marks as well as RNA-seq data. A predicted lncRNA of 72 kb is indicated in magenta.

DOI: https://doi.org/10.7554/eLife.29391.020

The following figure supplements are available for figure 10:

**Figure supplement 1.**
DOI: https://doi.org/10.7554/eLife.29391.021
**Figure supplement 2.**
DOI: https://doi.org/10.7554/eLife.29391.022
**Figure supplement 3.**
DOI: https://doi.org/10.7554/eLife.29391.023
**Figure supplement 4.** *T. gondii* peri-centromeric regions.
DOI: https://doi.org/10.7554/eLife.29391.024
**Figure supplement 5.** H4K31me1 marks long non-coding RNAs.
DOI: https://doi.org/10.7554/eLife.29391.025

buffer pH 7.2, 2 mM L-glutamine and 50 µg ml of penicillin and streptomycin (Invitrogen). Cells were incubated at 37°C in 5% $CO_2$. Type I (RH wildtype and RH $\Delta ku80$) and type II strains (Pru $\Delta ku80$) of *T. gondii* were maintained in vitro by serial passage on monolayers of HFFs. *P. falciparum* 3D7 strain was grown in RPMI 1640 media supplemented with 0.5% Albumax II, 0.1 mM hypoxanthine and 10 mcg/ml gentamicin. The culture was maintained at 2% hematocrit and 5% parasitemia. The parasites were grown at 37°C and in 1% $O_2$, 5% $CO_2$% and 94% $N_2$ gas mixture concentration. The cultures were free of mycoplasma, as determined by qualitative PCR.

## Immunofluorescence microscopy

*T. gondii*-infected HFF cells grown on coverslips were fixed in 3% formaldehyde for 20 min at room temperature, permeabilized with 0.1% (v/v) Triton X-100 for 15 min and blocked in phosphate-buffered saline (PBS) containing 3% (w/v) bovine serum albumin (BSA). The cells were then incubated for 1 hr with primary antibodies followed by the addition of secondary antibodies conjugated to Alexa Fluor 488 or 594 (Molecular Probes). Nuclei were stained for 10 min at room temperature with Hoechst 33258. Coverslips were mounted on a glass slide with Mowiol mounting medium, images were acquired with a fluorescence ZEISS ApoTome.2 microscope and images were processed by ZEN software (Zeiss). *P. falciparum* asexual blood life stages were washed with PBS and fixed in solution with 4% paraformaldehyde and 0.0075% glutaraldehyde in PBS for 30 min. After one wash with PBS, cells were permeabilized with 0.1% Triton X-100 in PBS for 10 min. Cells were washed twice with PBS, and blocked with 3% BSA in PBS for 1 hr. The cells were then incubated for 1 hr with primary antibodies followed by the addition of secondary antibodies conjugated to Alexa Fluor 488 or 594 (Molecular Probes). Nuclei were stained for 30 min at room temperature with Hoechst 33258. The parasites were finally washed 2–3 times before loading onto glass slides mixed with fluoro-gel (Electron Microscopy Sciences). Images were acquired with a fluorescence ZEISS ApoTome.2 microscope and processed by ZEN software (Zeiss).

## HDACi treatments

The final concentration of histone deacetylase inhibitors dissolved in DMSO was, as described (*Bougdour et al., 2013*), FR-235222 (90 nM), apicidin (100 nM), HC-toxin (100 nM), trichostatin A (100 nM), scriptaid (100 nM), APHA (100 mM) and sodium butyrate (5 mM). They were added to infected HFF cells for 18 hr. Halofuginone (10 nM) was shown to inhibit prolyl-tRNA synthetase (*Jain et al., 2015*) and was used as a control.

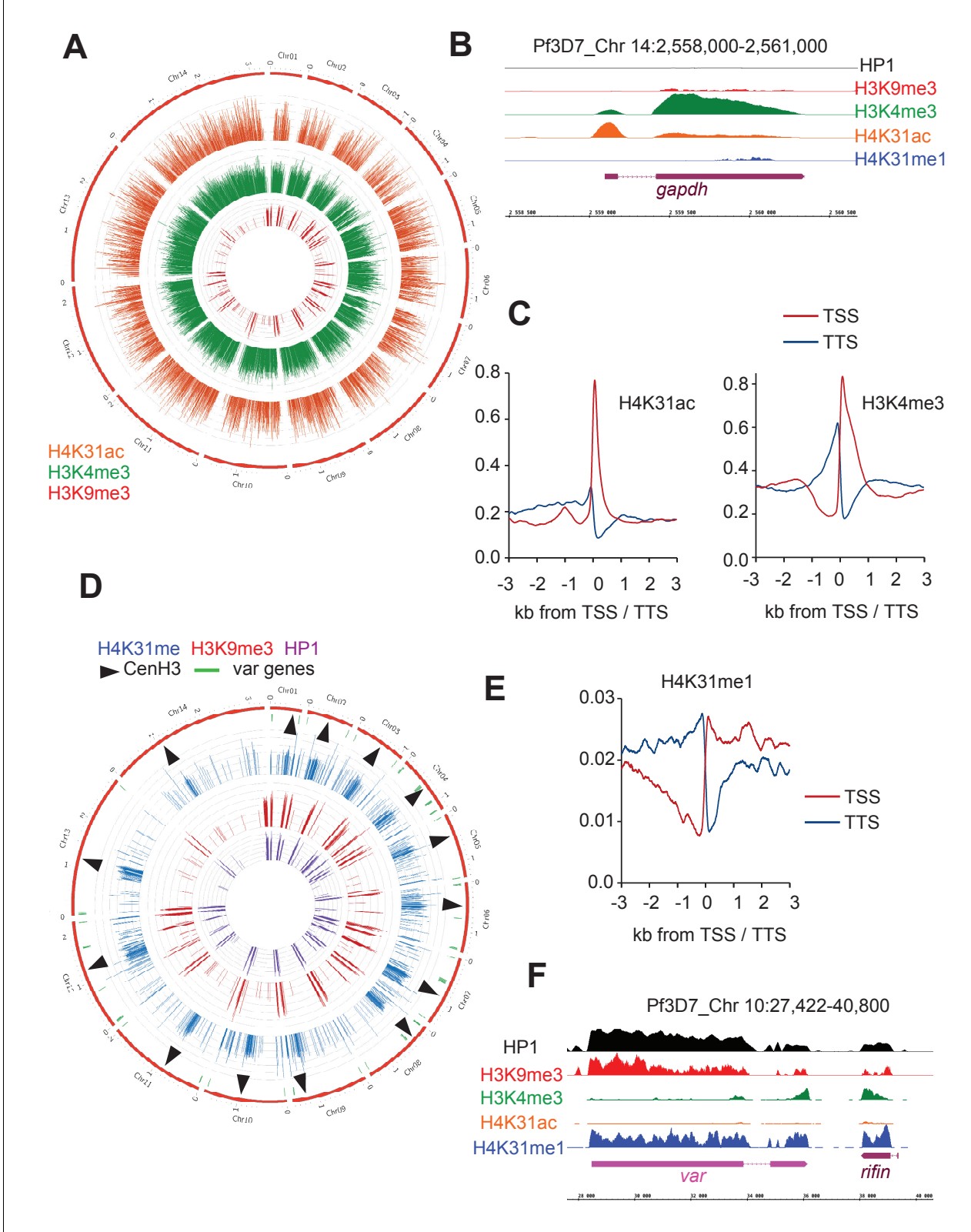

**Figure 11.** Genome-wide analysis of H4K31ac and H4K31me1 chromatin occupancy in *P. falciparum*. (a) Chromosomal projection of H4K31ac, H3K9me3 and H3K4me3 occupancies in *P. falciparum*. The full set of chromosomes is represented as the circular plot. (b) Zoomed-in view of PTMs and HP1 enrichment along the *gapdh* locus. (c) Plot of genome-wide H4K31ac and H3K4me3 occupancy profiles at peri-TSS (Transcription Start Sites) and –TTS (Transcription Termination Sites) regions. (d) Chromosomal projection of H4K31me1, H3K9me3 and HP1 occupancies in *P. falciparum*. The full set of

*Figure 11 continued on next page*

*Figure 11 continued*

chromosomes is represented as the circular plot where CenH3 locations (black arrow) and *var* genes (in green) are indicated. (**e**) Plot of genome-wide H4K31me1 occupancy profiles at peri-TSS and –TTS regions. (**f**) IGB view of a section of chromosome 10 showing enrichment of H4K31me1, H3K9me3 and HP1 at *var* gene.

DOI: https://doi.org/10.7554/eLife.29391.026

The following figure supplements are available for figure 11:

**Figure supplement 1.** Correlation matrix between *P. falciparum* ChIP-seq experiments.

DOI: https://doi.org/10.7554/eLife.29391.027

**Figure supplement 2.** Chromosomal projection of H4K31ac, H3K9me3 and H3K4me3 occupancies in *P. falciparum*.

DOI: https://doi.org/10.7554/eLife.29391.028

**Figure supplement 3.** Chromosomal projection of H4K31me1, H3K9me3 and HP1 occupancies in *P. falciparum*.

DOI: https://doi.org/10.7554/eLife.29391.029

## Plasmid constructs

To construct the vector pLIC-ENO1-HAFlag, the coding sequence of ENO1(TGME49_268860) was amplified using primers LIC-268860_Fwd (TACTTCCAATCCAATTTAGCgaacatgcagg-caatggcttggctcttc) and LIC-268860_Rev (TCCTCCACTTCCAATTTTAGC ttttgggtgtcgaaagctctctcccgcg) using Pruku80 genomic DNA as template. The resulting PCR product was cloned into the pLIC-HF-dhfr vector using the ligation independent cloning (LIC) method as reported previously (*Bougdour et al., 2013*).

## Cas9-mediated gene disruption in *T. gondii*

The plasmid pTOXO_Cas9-CRISPR was described by *Sangaré et al. (2016)*. For gene disruption using the CRISPR/Cas9 system, the genes of interests (GOI) were: GCN5A (*TGGT1_254555*), GCN5B (*TGGT1_243440*), MYST-A (*TGGT1_318330*), MYST-B (*TGGT1_207080*), HAT1 (*TGGT1_293380*), HDAC1 (*TGGT1_281420*), HDAC2 (*TGGT1_249620*), HDAC3 (*TGGT1_227290*), HDAC4 (*TGGT1_257790*) and HDAC5 (*TGGT1_202230*). Twenty mers-oligonucleotides corresponding to specific GOI were cloned using the Golden Gate strategy. Briefly, primers TgGOI-CRISP_FWD and TgGOI-CRISP_REV containing the sgRNA targeting TgGOI genomic sequence were phosphorylated, annealed and ligated into the linearized pTOXO_Cas9-CRISPR plasmid with BsaI, leading to pTOXO_Cas9-CRISPR::sgTgGOI. *T. gondii* tachyzoites were then transfected with the plasmid and grown on HFF cells for 18–36 hr. Cloning oligonucleotides used in this study were:

TgHDAC1-CRISP-FWD: 5'- AAGTTGCGTCGCCGTTCTCTCACGCG −3'
TgHDAC1-CRISP-REV: 5'- AAAACGCGTGAGAGAACGGCGACGCA −3'
TgHDAC2-CRISP-FWD: 5'- AAGTTGCGCCCGTCGCCTCCCCCGCG −3'
TgHDAC2-CRISP-REV: 5'- AAAACGCGGGGGAGGCGACGGGCGCA −3'
TgHDAC3-CRISP-FWD: 5'- AAGTTGATATCGGAAGTTACTACTAG −3'
TgHDAC3-CRISP-REV: 5'- AAAACTAGTAGTAACTTCCGATATCA −3'
TgHDAC4-CRISP-FWD: 5'- AAGTTGCTGTTGCTGAAGCCCAGGCG −3'
TgHDAC4-CRISP-REV: 5'- AAAACGCCTGGGCTTCAGCAACAGCA −3'
TgHDAC5-CRISP-FWD: 5'- AAGTTGGCGAGACCGGGGCAGCCGCG −3'
TgHDAC5-CRISP-REV: 5'- AAAACGCGGCTGCCCCGGTCTCGCCA −3'
TgGCN5A-CRISP-FWD: 5'- AAGTTGCGTGACGAACGACAGGCAAG −3'
TgGCN5A-CRISP-REV: 5'- AAAACTTGCCTGTCGTTCGTCACGCA −3'
TgGCN5B-CRISP-FWD: 5'- AAGTTGGGTTTCCTGTGTCGAGACCG −3'
TgGCN5B-CRISP-REV: 5'- AAAACGGTCTCGACACAGGAAACCCA −3'
TgMYSTA-CRISP-FWD: 5'- AAGTTGGCTGCTCCGCGACTCAGCGG −3'
TgMYSTA-CRISP-REV: 5'- AAAACCGCTGAGTCGCGGAGCAGCCA −3'
TgMYSTB-CRISP-FWD: 5'- AAGTTGCGCGAAGAAGGGAGAGAGCG −3'
TgMYSTB-CRISP-REV: 5'- AAAACGCTCTCTCCCTTCTTCGCGCA −3'
TgHAT1-CRISP-FWD: 5'- AAGTTGCCGACGGGTCACGGAGACTG −3'
TgHAT1-CRISP-REV: 5'- AAAACAGTCTCCGTGACCCGTCGGCA −3'

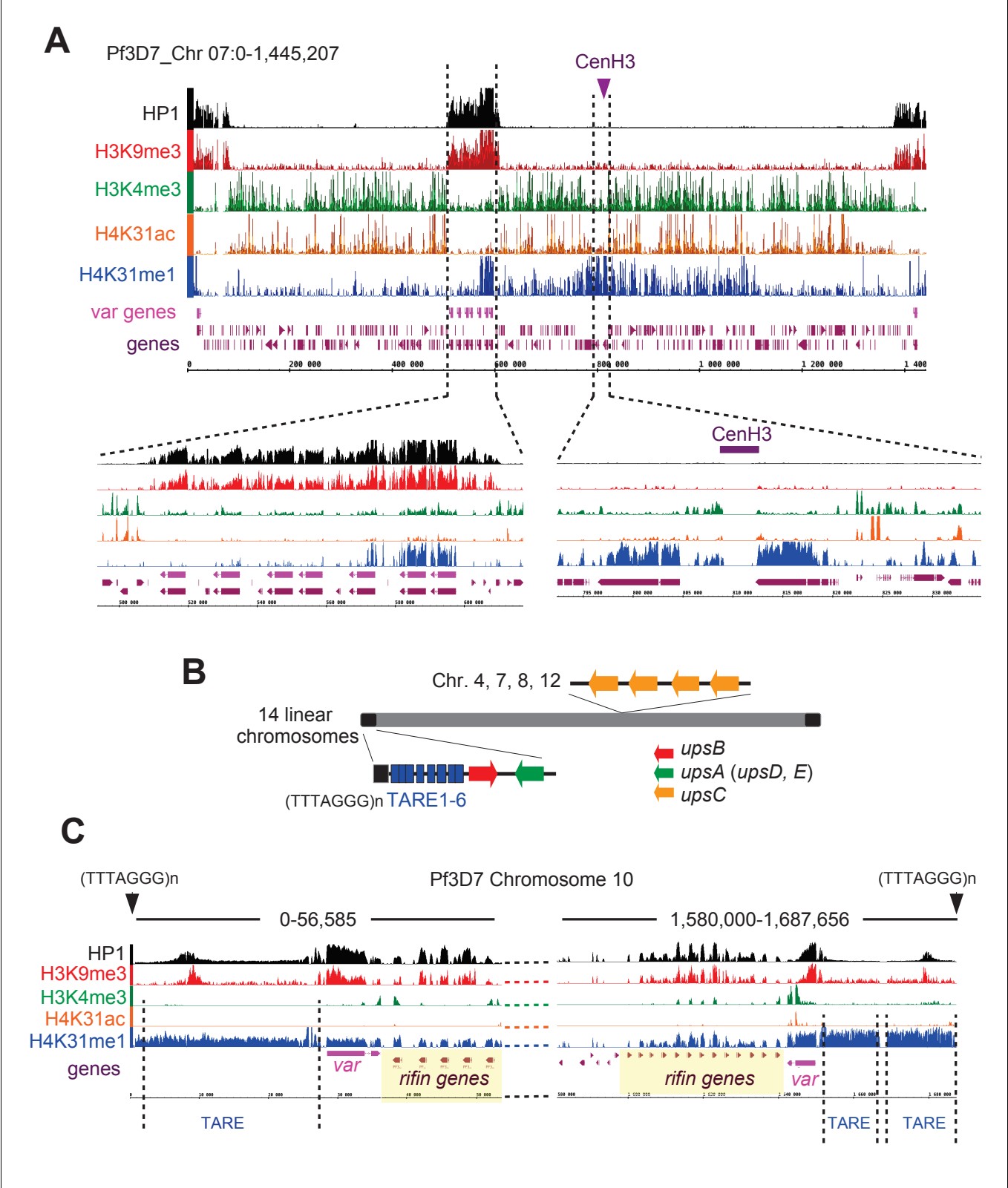

**Figure 12.** H4K31me1 singularly marks peri-centromeric heterochromatin in *P. falciparum.* (a) (Top) Chromosome-wide coverage plot of histone modifications and PfHP1 on *P. falciparum* Chr. 7. CenH3 was mapped according to *Hoeijmakers et al. (2012)* and *var* genes are indicated. (Bottom) Zoomed-in views of the *var*-genes-containing internal locus (left) and the centromeric region (right) from *P. falciparum* Chr. 7. (b) Genomic organization

*Figure 12 continued*

and nuclear position of *var* genes and telomere-associated repeat elements (TAREs) in *P. falciparum*. (c) IGB screenshots of *P. falciparum* sub-telomeric regions of chromosome 10. *rifin* and *var* genes as well as TAREs are highlighted.

DOI: https://doi.org/10.7554/eLife.29391.030

The following figure supplements are available for figure 12:

**Figure supplement 1.** H4K31me1 marks pericentromeric chromatin in *P. falciparum*.
DOI: https://doi.org/10.7554/eLife.29391.031

**Figure supplement 2.** H4K31me1 marks pericentromeric chromatin in *P. falciparum*.
DOI: https://doi.org/10.7554/eLife.29391.032

## *T. gondii* transfection

*T. gondii* RH, RH $\Delta ku80$ and Pru $\Delta ku80$ were electroporated with vectors in cytomix buffer (120 mM KCl, 0.15 mM CaCl$_2$, 10 mM K$_2$HPO$_4$/ KH$_2$PO$_4$ pH7.6, 25 mM HEPES pH7.6, 2 mM EGTA, 5 mM MgCl$_2$) using a BTX ECM 630 machine (Harvard Apparatus). Electroporation was performed in a 2 mm cuvette at 1.100V, 25 $\Omega$ and 25 $\mu$F. Stable transgenic parasites were selected with 1 $\mu$M pyrimethamine, single-cloned in 96 well plates by limiting dilution and verified by immunofluorescence assay.

## Antibodies

Primary antibodies: rabbit homemade anti-TgHDAC3 described in *Bougdour et al. (2009)*; RRID: AB_2713903), mouse anti-HA (Roche, RRID:AB_2314622), rabbit anti-H4K8ac (Millipore, RRID:AB_310524), rabbit anti-H4K12ac (Millipore, RRID:AB_11215637), rabbit anti-H3K4ac (Diagenode, RRID:AB_2713904), rabbit anti-H3K9ac (Diagenode RRID:AB_2713905), rabbit anti-H3K14ac (Diagenode, RRID:AB_2713906), rabbit anti-H3K18ac (Diagenode, RRID:AB_2713907), rabbit anti-H3K14ac (Millipore, RRID:AB_1977241) and mouse anti-H3K27ac (Diagenode, RRID:AB_2713908), anti-H4K20me3 (Diagenode, RRID:AB_2713909), anti-H3K9me3 (Millipore, RRID:AB_916348), anti-H3K4me1 (Diagenode, RRID:AB_2637078) and anti-H3K4me3 (Diagenode, RRID:AB_2616052). Western blot secondary antibodies were conjugated to alkaline phosphatase (Promega), whereas immunofluorescence secondary antibodies were coupled with Alexa Fluor 488 or Alexa Fluor 594 (Thermo Fisher Scientific). We also raised homemade H4K31acetylation- and H4K31monomethylation-specific antibodies in rabbit against linear peptides corresponding to amino acid residues 23–35 of histone H4 and carrying modified residue K31: C-DNIQGIT<u>K</u>me1PAIR; C-DNIQGIT<u>K</u>acPAIR and C-<u>R</u>DNIQGIT<u>K</u>acPAIR. They were produced by Eurogentec and used for immunofluorescence, immunoblotting and chromatin immunoprecipitation.

## Histone purification-, immunoblotting- and mass spectrometry-based proteomic analysis

For histone purification, HFF cells were grown to confluence and infected with Pru$\Delta ku80$ parasites. Intracellular tachyzoites were treated with histone deacetylase HDAC3 inhibitor, 90 nM FR235222 for 18 hr. As an appropriate control, we treated tachyzoites with 0.1% DMSO. Histones were extracted and purified using histone purification kit (Active motif) according to manufacturer's protocol. For western blotting, histone proteins were run on a *NuPAGE* 4–12% Bis-Tris polyacrylamide gels in MES-SDS running buffer (Invitrogen) and transferred to a polyvinylidene fluoride PVDF membrane (Immobilon-P; Millipore) using NuPAGE transfer buffer (Invitrogen). The blots were probed using primary antibodies: pan acetyl H4, H4K31ac and H4K31me1, followed by phosphatase-conjugated goat secondary antibodies (Promega). The expected band of histones were detected using NBT-BCIP (Amresco). Nucleosomes from *T. gondii*-infected cells were purified and proteins separated by SDS-PAGE. The band corresponding to H4 was excised and its protein content digested using trypsin. The resulting peptides were submitted to mass spectrometry-based proteomic analysis (U3000 RSLCnano coupled to Q-Exactive HF, Thermo Scientific). Peptides and proteins were identified using Mascot software (Matrix Science).

## Immunodot blot assay

The MODifiedTM Histone Peptide Array (Activemotif) contains 59 different post-translational modifications for histone acetylation, methylation, phosphorylation and citrullination on the N-terminal tails of histones H2A, H2B, H3 and H4. Each 19mer peptide may contain up to four modifications each. Five control spots are included on each array: biotin peptide, c-Myc tag, no histone peptide and two background spots containing a mixture of modifications that are present on the array. Arrays were blocked with tris-buffered saline (TTBS) (10 mM Tris [pH 7.4], 150 mM NaCl, 0.05% Tween 20) plus 5% nonfat dry milk. Antibodies were diluted in TTBS. Primary antibodies were detected using horse-radish peroxidase (HRP)-conjugated anti-IgG antibodies (R and D systems). The blots were developed with the SuperSignal West Pico Chemiluminescent Substrate kit (Thermo Fisher Scientic). The Array Analyze software program designed by Activemotif was used to analyze the spot intensity of the interactions and to generate graphical analysis of the histone peptide–antibody interactions. The Spot Statistics tab gives a comparison of the intensities of each spot in the left and right array. The data points ideally should be on a straight line connecting 0,0 and 1,1 which indicates perfect duplicates. Outliers (or poor replicates) will not fall along the linear line. The Distribution Errors tab displays the errors of the intensities between the right and left spot, normalized to the maximum intensity. Ideally, most of the peptides will contain an error range of 0–5% (which means very little variation between the left and right sides). The Reactivity tab displays the background-subtracted intensity values for all single modified peptides, as well as for control spots. Selecting a modification from the pull-down menu scales the intensity of the selected modification to 1.0 (y-axis). The impact of neighboring modifications on the single modified peptides was further investigated using the Modification Analysis tab and selecting for the modification of interest. The results were graphed as a Specificity factor, which is the ratio of the average intensity of all spots containing the mark divided by the intensity of all spots not containing the mark. All modifications have been accounted for in determining the specificity factor, but only the ten modifications with the highest specificity factor values are graphed. The larger the difference in specificity-factor values for the mark of interest versus other marks, the more specific the interaction. Non-specific signals will decrease the specificity-factor values.

## Chromatin Immunoprecipitation and next -generation sequencing in *T. gondii*

HFF cells were grown to confluence and infected with the type II (PruΔ*ku80*) strain. Harvested intracellular parasites were crosslinked with formaldehyde (final concentration 1%) for 8 min at room temperature and the crosslinking was stopped by the addition of glycine (final concentration 0.125M) for 5 min at room temperature. Crosslinked chromatin was lysed in ice-cold lysis buffer (50 mM HEPES KOH pH7.5, 140 mM NaCl, 1 mM EDTA, 10% glycerol, 0.5% NP-40, 0.125% Triton X-100, protease inhibitor cocktail) and sheared in shearing buffer (1 mM EDTA pH8.0, 0.5 mM EGTA pH8.0, 10 mM Tris pH8.0, protease inhibitor cocktail) by sonication using a Diagenode Biorupter. Samples were sonicated for 16 cycles (30 s ON and 30 s OFF), to 200–500 base-pair average size. Immuno-precipitation was carried out using sheared chromatin, 5% BSA, protease inhibitor cocktail, 10% Triton X-100, 10% deoxycholate, DiaMag Protein A-coated magnetic beads (Diagenode) and antibodies (H4K31ac, H4K31me1, pan acetyl H4, H4K20me3, H3K9me3, H3K4me3, H3K4me1 and H3K14ac). A rabbit IgG antiserum was used as a control mock. After overnight incubation at 4°C on rotating wheel, chromatin-antibody complexes were washed and eluted from beads by using iDeal ChIP-seq kit for Histones (Diagenode) according to the manufacturer's protocol. Samples were decrosslinked by heating for 4 hr at 65°C. DNA was purified by using the IPure kit (Diagenode) and quantified using Qubit Assays (Thermo Fisher Scientific) according to the manufacturer's protocol. For ChIP-seq, purified DNA was used to prepare libraries and then sequenced by Arraystar (USA, http://www.arraystar.com/).

## Library preparation, sequencing and data analysis (Arraystar)

ChIP-Sequencing library preparation was performed according to Illumina's protocol Preparing Samples for ChIP Sequencing of DNA. For library preparation: 10 ng DNA of each sample was converted to phosphorylated blunt-ended with T4 DNA polymerase, Klenow polymerase and T4 polymerase (NEB). An 'A' base was added to the 3' end of the blunt phosphorylated DNA fragments using the

polymerase activity of Klenow (exo minus) polymerase (NEB). Illumina's genomic adapters were ligated to the A tailed DNA fragments, and PCR amplification was performed to enrich ligated fragments using Phusion High Fidelity PCR Master Mix with HF Buffer (Finnzymes Oy). The enriched product of ~200–700 bp was cut out from the gel and purified. For sequencing, the library was denatured with 0.1M NaOH to generate single-stranded DNA molecules, and loaded onto channels of the flow cell at 8pM concentration, amplified in situ using the TruSeq Rapid SR cluster kit (#GD-402–4001, Illumina). Sequencing was carried out by running 100 cycles on Illumina HiSeq 4000 according to the manufacturer's instructions. For data analysis, after the sequencing platform generated the sequencing images, the stages of image analysis and base calling were performed using Off-Line Basecaller software (OLB V1.8). After passing the Solexa CHASTITY quality filter, the clean reads were aligned to the *T. gondii* reference genome (Tgo) using BOWTIE (V2.1.0). Aligned reads were used for peak calling of the ChIP regions using MACS V1.4.0. Statistically significant ChIP-enriched regions (peaks) were identified by comparison of two samples, using a p-value threshold of $10^{-5}$. The peaks in each sample were then annotated by the overlapped gene using the newest *T. gondii* database. The EXCEL/BED format file containing the ChIP-enriched regions was generated for each sample. For data visualization, the mapped 100 bp reads represent enriched DNA fragments identified by the ChIP experiment. Any region of interest in the raw ChIP-seq signal profile can be directly visualized in the genome browser. We used 10-bp resolution intervals (10-bp bins) to partition the stacked reads region, and counted the number of reads in each bin. All of the 10-bp resolution ChIP-seq profiles of each sample are saved as UCSC wig format files, which can be visualized in the *T. gondii* Genome Browser (http://protists.ensembl.org/Toxoplasma_gondii/Info/ Index). All of these raw and processed files can be found at Series GSE98806.

## Chromatin immunoprecipitation and next-generation sequencing in *P. falciparum*

Chromatin from synchronous-rings-stage parasites of 3D7 clone G7 was prepared and $3*10^8$ cells per ChIP used for the previously described protocol (*Lopez-Rubio et al., 2013*). Briefly, chromatin was crosslinked in 1% formaldehyde for 10 min (Sigma-Aldrich, #SZBD1830V), sheared to an average length of 300 bp using the BioRuptor Pico, and individual histone modifications were pulled down using 0.5 µg of antibody for H3K4me3 (Diagenode, cat # K2921004), H3K9me3 (Millipore, cat # 257833), and homemade rabbit polyclonal anti-PfHP1. 5 µl rabbit polyclonal anti-H4K31me1 and 15 µl anti-H4K31ac were used for each experiment. To generate Illumina-compatible sequencing libraries, the immunoprecipitated DNA and input was processed using the MicroPlex Library Preparation Kit (Diagenode C05010014) according to manufacturer's instructions. The optimized library amplification step used KAPA Biosystems HIFI polymerase (KAPA Biosystems KK2101). Pooled, multiplexed libraries were sequenced on an Illumina NextSeq 500/550 system as a 150-nucleotide single-end run. The raw data were demultiplexed using bcl2fastq2 (Illumina) and converted to fastq format files for downstream analysis. Two biological replicates were analyzed for each antibody.

## *P. falciparum* ChIP-seq data analysis

Sequencing reads were mapped to the *P. falciparum* 3D7 genome assembly (PlasmoDB v3.0) with the Burrows-Wheeler Alignment tool (BWA) using default settings, and then sequences were quality filtered by the Q20 Phred quality score. ChIP-seq peak calling was performed using the MACS2 algorithm. For genome-wide representation of each histone mark's distribution, the coverage was calculated as average reads per million over bins of 1000 nucleotides using bamCoverage from the package deepTools. Correlations of the different biological replicates were calculated by performing Pearson's and Spearman's correlation analysis of pairwise comparisons of BAM alignment files, and ChIP-seq peak enrichment scores (log2) using the MACS2 and deepTools. Circular and linear coverage plots were generated using Circos and Integrated Genomics Viewer, respectively. All of the raw and processed files can be found at NCBI Bioproject ID PRJNA386433.

## Acknowledgements

This work was supported by the Agence Nationale Recherche, France (grant ANR Blanc 2012 TOXOHDAC, grant number ANR-12-BSV3-0009-01), the Laboratoire d'Excellence ParaFrap, France (grant number ANR-11-LABX-0024]), the Atip-Avenir and Finovi programs (Apicolipid projects to

CYB); the European Research Council (ERC Consolidator Grant N° 614880 Hosting TOXO to MAH and ERC AdG N° 670301 PlasmoSilencing to AS). Proteomic experiments were partly supported by the ProFi grant (ANR-10-INBS-08–01).

## Additional information

### Funding

| Funder | Grant reference number | Author |
|---|---|---|
| European Commission | ERC Consolidator Grant No. 614880 | Fabien Sindikubwabo Dominique Cannella Mohamed-ali Hakimi |
| Agence Nationale de la Recherche | LABEX PARAFRAP ANR-11-LABX-0024 | Fabien Sindikubwabo Shuai Ding Tahir Hussain Dominique Cannella Andrés Palencia Alexandre Bougdour Artur Scherf Mohamed-ali Hakimi Cyrille Y Botté |
| Agence Nationale de la Recherche | ANR-12-BSV3-0009-01 | Dominique Cannella Andrés Palencia Alexandre Bougdour Mohamed-ali Hakimi |
| Agence Nationale de la Recherche | ANR-10-INBS-08-01 | Lucid Belmudes Yohann Couté |
| CNRS-INSERM-Fondation FINOVI | Atip-Avenir Program Project Apicolipid | Cyrille Y Botté |
| Agence Nationale de la Recherche | ANR-12-PDOC-0028 | Cyrille Y Botté |
| European Commission | ERC AdG No. 670301 | Artur Scherf |

The funders had no role in study design, data collection and interpretation, or the decision to submit the work for publication.

### Author contributions

Fabien Sindikubwabo, Conceptualization, Resources, Formal analysis, Validation, Investigation, Visualization, Methodology; Shuai Ding, Resources, Investigation, Visualization, Methodology; Tahir Hussain, Alexandre Bougdour, Resources, Investigation, Visualization; Philippe Ortet, Mohamed Barakat, Data curation, Software, Formal analysis, Visualization; Sebastian Baumgarten, Resources, Data curation, Software, Formal analysis, Visualization; Dominique Cannella, Resources, Investigation; Andrés Palencia, Investigation, Visualization; Lucid Belmudes, Formal analysis, Investigation, Methodology; Yohann Couté, Conceptualization, Investigation, Visualization; Isabelle Tardieux, Conceptualization, Writing—original draft, Writing—review and editing; Cyrille Y Botté, Conceptualization, Investigation; Artur Scherf, Conceptualization, Supervision, Funding acquisition, Writing—review and editing; Mohamed-ali Hakimi, Conceptualization, Formal analysis, Supervision, Funding acquisition, Investigation, Methodology, Writing—original draft, Project administration, Writing—review and editing

### Author ORCIDs

Andrés Palencia http://orcid.org/0000-0002-1805-319X
Yohann Couté http://orcid.org/0000-0003-3896-6196
Mohamed-ali Hakimi http://orcid.org/0000-0002-2547-8233

### Decision letter and Author response

Decision letter https://doi.org/10.7554/eLife.29391.039
Author response https://doi.org/10.7554/eLife.29391.040

## Additional files

### Supplementary files

• Transparent reporting form
DOI: https://doi.org/10.7554/eLife.29391.033

### Major datasets

The following datasets were generated:

| Author(s) | Year | Dataset title | Dataset URL | Database, license, and accessibility information |
|---|---|---|---|---|
| Sindikubwabo F, Ding S, Hussain T, Ortet P, Barakat M, Baumgarten S, Cannella D, Palencia A, Bougdour A, Belmudes L, Coute Y, Tardieux I, Botte CY, Scherf A, Hakimi MA | 2017 | Chromosomal organization and genic expression in Apicomplexan parasites is critically dictated by a versatile acetylation-methylation switch at K31 on the lateral surface of histone H4 | https://www.ncbi.nlm.nih.gov/geo/query/acc.cgi?acc=GSE98806 | Publicly available at the NCBI Gene Expression Omnibus (accession no: GSE98806) |
| Sindikubwabo F, Ding S, Hussain T, Ortet P, Barakat M, Baumgarten S, Cannella D, Palencia A, Bougdour A, Belmudes L, Coute Y, Tardieux I, Botte CY, Scherf A, Hakimi MA | 2017 | Plasmodium falciparum histone post-translational modifications | https://www.ncbi.nlm.nih.gov/bioproject/?term=PRJNA386433 | Publicly available at the NCBI BioProject database (accession no: PRJNA386433) |

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
