## [Decision Letter]

Thank you for submitting your article "Modifications at K31 on the lateral surface of histone H4 contribute to genome structure and expression in apicomplexan parasites" for consideration by *eLife*. Your article has been reviewed by three peer reviewers, one of whom is a member of our Board of Reviewing Editors, and the evaluation has been overseen by Wendy Garrett as the Senior Editor.

The reviewers have discussed the reviews with one another and the Reviewing Editor has assembled a list of concerns that we would like you to address in a response letter.

Summary:

This paper describes the characterization of histone post-translational modifications H4K31Ac and H4K31me1 in the apicomplexan parasites *P. falciparum* and *T. gondii*. Most characterized histone PTMs in these organisms have been within the N terminal tails and globular domain PTMs are a recent focus of study in the chromatin literature. The authors propose a PTM switch on H4K31 that governs chromatin state in *T. gondii*. The authors also identify the enzyme responsible for addition and removal of H4K31Ac. The modifications described here have not been examined closely in any organism, and the data presented suggest potential for important functions in gene regulation and chromosome function. Data are also presented suggesting a function for H4K31me1 in metazoan mitosis. Overall the findings appear interesting and potentially important, but there are a number of methodological points that must be clarified and analyses that must be supplied. Aspects of data presentation and literature citation also require improvement.

Essential revisions:

1) The specificity of the H4K31Ac and H4K31me1 antibodies is crucial, but these data aren't shown. Modification antibodies that are truly specific are difficult to generate, and there have been many reports of commercial H4ac and histone lysine methyl antibodies cross-reacting with other modified residues or reactivity influenced by other nearby PTMs (Rothbart 2015 Mol Cell). Much of the data on H4K31Ac is consistent with the antibody having cross-reactivity to H3K14Ac. Please show antibody specificity data, which should demonstrate that the antibodies do not cross-react with common histone PTMs, perhaps by probing available PTM arrays. This is the most essential revision and the authors should insure that antibody specificity is unambiguously demonstrated.

2) Figure 1 shows the MS/MS analysis detecting H4K31ac but it is not clear from the text how this MS was performed (Results section and Materials and methods say Tg infected cells) and how extensively the PTM on histones or H4 were examined. The sample size disclosure form says 3 replicates were performed, but we are not given any information about the concordance of these replicates or full data. Did the authors detect the modification in Plasmodium by MS? The modifications were not reported in the cited Trelle 2009 paper. If the authors did not perform MS for Pf H4, they should cite Cobbold 2016 Science Reports who found H4K31ac using an affinity approach similar to the one performed by Jeffers 2012 for the Tg acetylome, and Saraf 2016 JProtRes, which is in the references, but not cited as having reported this PTM (Introduction, last paragraph).

3) More conclusive evidence for the H4K31me1 PTM in Tg and/or Pf must be provided, as the authors did not cite a reference nor could we find published data that such a PTM has been reported using MS. Did the authors detect H4K31me1 in their 3 MS replicates? As cited by the authors, Nardelli (mBIO 2013) detected H4K31succ but the authors do not mention 2 additional modifications, H4K31me2 and H4K31form, that were also reported in that study. Based on the genome localization the abundance of the PTM appears greater than H4K31ac. Further, given the access to specific antibodies, an affinity approach should be straightforward to confirm the PTM even if the bulk histone approach was unsuccessful.

4) Conclusions derived from cytological data are usually supported by just one image, or at best several images. The claims regarding the mitotic role of H4K31me1 in metazoans (paragraph four in subsection "H4K31me1 associates in vivo with distinct chromatin patterns") are very weakly supported. The conclusion stated in paragraph three is not supported at all. A quantitative analysis of the cytology should be presented (percentages of images matching those shown) with statistics where appropriate. All conclusions should be supported by data, as in:

a) Figure 1 FR235222 was reported previously by the authors to be a specific inhibitor of HDAC3. The image provided suggests that this inhibitor, but not other HDAC inhibitors, change the abundance of H4K31ac. Quantitation and appropriate statistical analysis is required.

b) Figure 2 Pf parasites are shown in similar experiments as Figure 1. These experiments also require quantitation.

c) Figure 3 CRISPR inactivation of candidate TgHDAC was performed. Again, the images shown are convincing, but single images are not sufficient. The authors need to quantify +red/green signals and provide statistical analysis.

d) Figure 4 CRISPR inactivation of candidate HAT. The authors need to quantify +red/green signals and provide statistical analysis.

e) The authors state that cyclopeptide inhibitors impact H4K31Ac levels and that other types of HDAC inhibitors do not. However, from Figure 3, it appears that butyrate and APHA influence H4K31Ac levels.

5) There are several references to low levels of H4K31ac, but these are not well-supported, and appear to be derived from subjective interpretation of cytology (comparing signal intensity across antibodies is not appropriate). Perhaps mass spectroscopy provides a more solid basis for this assertion, but mass spec data are presented very briefly in Figure 1.

6) The presentation of genomics data relies too heavily on genome browser snapshots. These are helpful and appropriate as supporting evidence, but conclusions based on genomic data should be supported by genomic analyses. Some conclusions, like the limitation of H4K31me1 by translational initiation and termination sites (subsection "Nucleosomes with H4K31ac and H4K31me1 are enriched at the promoter and the core body of active genes, respectively") are not supported at all – transcriptional initiation and termination sites could just as easily be involved. The claim that H3K9me3 and H3K14ac co-localize in poorly transcribed (Q4) genes (Figure 8) is inconsistent with the genomic analysis in Figure 7—figure supplement 1, which does not show an obvious enrichment of either modification in Q4 genes. This may be because Q4 genes are transcriptionally heterogeneous (Figure 7).

7) Figure 6 and subsection "Nucleosomes with H4K31ac and H4K31me1 are enriched at the promoter and the core body of active genes, respectively" state that H4K31me1 is a novel mark of transcriptional units, but H3K4me1 (genome-wide data available on Toxodb) seems to have a very similar distribution (also suggested by H3K4me1 data in Figure 7—figure supplement 1, though not discussed), so this statement may not be accurate. Did the authors compare H3K4me1 and H3K31me1 distributions?

8) Figure 8 shows that H4K31me1 marks Tg pericentromeric heterochromatin (discussed in paragraph three of subsection "Distribution of H4K31 modifications across the *P. falciparum* genome reveals 403 H4K31me1 as a novel pericentromeric PTM"), but this information appears to be based upon the genome-wide data for CenH3 published in Brooks et al., 2013 (also available at www.Toxodb.org). The Brooks paper, which showed the localization of H3K9me2/3 to Tg centromeres should be discussed and cited since the authors examine similarities and differences between Tg and Pf. The Tg ortholog of HP1, which binds H3K9me3, was reported by Gissot et al., 2012 (also not cited), and its localization was reported to centromeres and telomeres.

9) The data reported for H3K9me3 in this manuscript agree in centromeric localization but conflict in reporting localization to "poised" genes. The definition of "poised" genes is unclear. A table is provided (without a legend or description of samples), but statistical support for enrichment of this mark on poised genes is required, as is further discussion. What proportion of genes in Q4 were poised and were they developmental stage-specific genes?

10) There are several species in which ncRNA contribute to the organization of centromeres (although probably not Tg based on the authors' data). The Scherf laboratory has characterized ncRNA in Plasmodium, including several that are transcribed at var loci. Are the H4K31me1 marks that flank centromeres in Pf sites of ncRNA transcription? Does the presence of ncRNA at var loci correlate with H4K31me1 localization?

11) The discussion of H4K31Ac and H4K31me1 functionality is overbroad. There are multiple functional claims: "a key role in gene regulation", "pivotal role...in pericentromeric heterochromatin", "regulating chromosomal condensation and segregation during mitosis", "H4K31me [...] substantially slows the progress of the RNA polymerase on active genes". This paper presents no direct functional data and hence cannot claim to demonstrate any role for H4K31Ac or H4K31me1. Reasonable inferences can be made based on genomic localization and correlations with other PTMs and gene expression, but it's important to be very clear that functional tests are yet to be done. This includes the title, which makes unsupported functional claims.

12) The discussion of H4K31Ac and H4K31me1 functionality is vague and the descriptions of the genomic localization of H4K31Ac and H4K31me1 are very difficult to follow. H4K31me1 may function by preventing acetylation (but not have a direct influence on nucleosomes), inhibiting histone exchange (or otherwise repressing transcription directly or through a reader), or both. The authors seems to favor the last possibility, but don't explain why.

13) The similarities and differences between *T. gondii* and *P. falciparum* need to be discussed more clearly. More generally, references to evolutionary relationships are confusingly vague. "Higher eukaryote" is an anachronistic term with little meaning; in most cases the authors appear to mean "animals". Chromatin organization in apicomplexa is described as unique, but many fungi have similar distributions of active and silent chromatin. *P. falciparum* and *T. gondii* are described as "closely linked" despite a divergence time of about 800 MY.

[Editors' note: further revisions were requested prior to acceptance, as described below.]

Thank you for submitting your article "Modifications at K31 on the lateral surface of histone H4 contribute to genome structure and expression in apicomplexan parasites" for consideration by *eLife*. Your article has been reviewed by two peer reviewers, and the evaluation has been overseen by a Reviewing Editor and Wendy Garrett as the Senior Editor. The reviewers have opted to remain anonymous.

The reviewers have discussed the reviews with one another and the Reviewing Editor has drafted this decision to help you prepare a revised submission.

Summary:

The reviewers and editors feel that the manuscript has been markedly improved and appreciate the extensive revisions carried out by the authors. However, we do have remaining substantial concerns in two areas.

Essential revisions:

1) Antibody specificity. The figure supplements for Figure 1 are difficult to interpret. The authors test the specificity of the antibodies using dot blots and modified histone peptide arrays. In these figures, part C shows a bar graph of a "specificity factor". The relevant figure legends direct the reader to the Materials and methods section, but we could not find an explanation of what a specificity factor is or how it is calculated in the Methods, and more generally the Materials and methods section does not explain how antibody specificity was determined. This needs to be explained so that the readers can understand whether the specificity factors for the H4 K31Ac antibody for modifications such as H3 T11P are significant. Part D of these figures presumably shows the quantitation of the binding of the antibody to the peptide arrays. However, again there is no explanation of how this data was derived or even what the Y-axis represents. This needs to be explained. Based on the data shown, it appears that the H4 K31 Ac antibody can bind to the unmodified H3 tail. If this is the case, the immunofluorescence data in the manuscript is impossible to interpret. Based on the tight cropping of the gels in Figure 1 it is not possible to tell whether the H4 K31Ac antibody recognizes H3. The full gels should be provided.

2) Unwarranted mechanistic claims. The manuscript still contains mechanistic claims about the function of modifications on H4 K31 that are speculative and not supported by data. The primary example of this is in the Abstract where the authors state "H4K31 acetylation promotes a relaxed chromatin state at the promoter of active genes through nucleosome disassembly in both parasites." The manuscript shows that H4 K31 Ac localization correlates with active chromatin but there is no data that shows that H4 K31 acetylation has a direct effect on chromatin structure through nucleosome disassembly. These types of claims should be removed.

---

## [Author Response]

Essential revisions:1) The specificity of the H4K31Ac and H4K31me1 antibodies is crucial, but these data aren't shown. Modification antibodies that are truly specific are difficult to generate, and there have been many reports of commercial H4ac and histone lysine methyl antibodies cross-reacting with other modified residues or reactivity influenced by other nearby PTMs (Rothbart 2015 Mol Cell). Much of the data on H4K31Ac is consistent with the antibody having cross-reactivity to H3K14Ac. Please show antibody specificity data, which should demonstrate that the antibodies do not cross-react with common histone PTMs, perhaps by probing available PTM arrays. This is the most essential revision and the authors should insure that antibody specificity is unambiguously demonstrated.

We performed an exhaustive characterization of both H4K31ac- and H4K31me1-directed antibodies using modified histone PTM arrays and added those data in Figure 1—figure supplement 1,Figure 1—figure supplement 2,Figure 1—figure supplement 3 and Figure 1—figure supplement 4. We have shown that our home-made antibody were highly specific for synthetic peptides with H4K31ac or H4K31me1 over peptides with previously described acetyl marks in histone tails and globular domains. H3K14ac and H4K20me1 were used as controls and showed no reactivity toward H4K31ac and H4K31me1 peptides, respectively. Taken together, these data provide compelling evidence that our antibodies recognize their respective epitopes with high specificity.

2) Figure 1 shows the MS/MS analysis detecting H4K31ac but it is not clear from the text how this MS was performed (Results section and Materials and methods say Tg infected cells) and how extensively the PTM on histones or H4 were examined. The sample size disclosure form says 3 replicates were performed, but we are not given any information about the concordance of these replicates or full data. Did the authors detect the modification in Plasmodium by MS? The modifications were not reported in the cited Trelle 2009 paper. If the authors did not perform MS for Pf H4, they should cite Cobbold 2016 Science Reports who found H4K31ac using an affinity approach similar to the one performed by Jeffers 2012 for the Tg acetylome, and Saraf 2016 JProtRes, which is in the references, but not cited as having reported this PTM (Introduction, last paragraph).

We did not perform ms-ms identification for Pf H4. However, H4K31ac was identified in Pf by two comprehensive studies of *P. falciparum* proteome and acetylome: Saraf et al., 2016 and Cobbold et al., 2016, respectively that we are citing properly in the revised manuscript (Results section, first paragraph). We performed several ms assays in *T. gondii* but modifications at this residue were quite difficult to identify with great confidence by mass spectrometry in our conditions. H4K31ac peptide was at least identified one time with high confidence (as reported in Figure 1). Using an affinity approach similar to Jeffers and Sullivan, 2012, similar peptides were detected by *Maxquant* and *Proline* but with low confidence. We did not mention those unsuccessful identifications and changed the sample size disclosure accordingly. Nevertheless, H4K31ac was distinctly identified in *T. gondii* by two independent studies (Jeffers and Sullivan, 2012; Xue et al., 2013) as well as in metazoans (Garcia et al., 2005; Garcia et al., 2006; Soldi et al., 2014). Taken together, these findings emphasize the existence H4K31ac on nucleosomes isolated from animal cells or *apicomplexa* parasites.

3) More conclusive evidence for the H4K31me1 PTM in Tg and/or Pf must be provided, as the authors did not cite a reference nor could we find published data that such a PTM has been reported using MS. Did the authors detect H4K31me1 in their 3 MS replicates? As cited by the authors, Nardelli (mBIO 2013) detected H4K31succ but the authors do not mention 2 additional modifications, H4K31me2 and H4K31form, that were also reported in that study. Based on the genome localization the abundance of the PTM appears greater than H4K31ac. Further, given the access to specific antibodies, an affinity approach should be straightforward to confirm the PTM even if the bulk histone approach was unsuccessful.

Identification of H4K31me1 using a bulk histone approach was unsuccessful in our conditions. Two other studies were also not able to identify the mark in the proteome of *T. gondii* (Nardelli et al., 2013) and *P. falciparum* (Saraf et al., 2016) while this PTM was identified with high confidence and several times in mammalian cells (Garcia et al., 2006; Moraes et al., 2015). We then used as an alternative approach antibodies directed specifically against methylated H4K31. As stated in subsection "H4K31me1 associates in vivo with distinct chromatin patterns", we raised an antibody directed against H4K31me2, a modification reported by Nardelli et al., 2013, however although reacting avidly with the peptide antigen, those did not detect histone H4 in human or parasite protein extracts. We also generated antibodies recognizing H4K31me1 and those were controlled for their specificy as above-mentioned.

We edited the manuscript accordingly specifically by mentioning all the other PTM occurring at H4K31; corresponding references were also added.

4) Conclusions derived from cytological data are usually supported by just one image, or at best several images. The claims regarding the mitotic role of H4K31me1 in metazoans (paragraph four in subsection "H4K31me1 associates in vivo with distinct chromatin patterns") are very weakly supported. The conclusion stated in paragraph three is not supported at all. A quantitative analysis of the cytology should be presented (percentages of images matching those shown) with statistics where appropriate. All conclusions should be supported by data, as in:

Quantitation and statistics were added in the new Figure 7.

a) Figure 1 FR235222 was reported previously by the authors to be a specific inhibitor of HDAC3. The image provided suggests that this inhibitor, but not other HDAC inhibitors, change the abundance of H4K31ac. Quantitation and appropriate statistical analysis is required.

Quantitation and statistics were added in the new Figure 2.

b) Figure 2 Pf parasites are shown in similar experiments as Figure 1. These experiments also require quantitation.

Quantitation and statistics were added in the new Figure 3.

c) Figure 3 CRISPR inactivation of candidate TgHDAC was performed. Again, the images shown are convincing, but single images are not sufficient. The authors need to quantify +red/green signals and provide statistical analysis.

Quantitation and statistics were added in the new Figure 4.

d) Figure 4 CRISPR inactivation of candidate HAT. The authors need to quantify +red/green signals and provide statistical analysis.

Quantitation and statistics were added in the new Figure 5.

e) The authors state that cyclopeptide inhibitors impact H4K31Ac levels and that other types of HDAC inhibitors do not. However, from Figure 3, it appears that butyrate and APHA influence H4K31Ac levels.

While we agree with the reviewer that both Butyrate and APHA trigger more acetylation of H4K31 in the host cell nuclei they fail to increase H4K31ac signal in parasite nuclei as seen in Figure 4 and quantified in new Figure 4.

5) There are several references to low levels of H4K31ac, but these are not well-supported, and appear to be derived from subjective interpretation of cytology (comparing signal intensity across antibodies is not appropriate). Perhaps mass spectroscopy provides a more solid basis for this assertion, but mass spec data are presented very briefly in Figure 1.

We agree with the reviewer comment that comparing signal intensity across antibodies may lead to subjective interpretations. Therefore and in absence of any quantitative ms-ms data, we have removed any mention to "low level" of H4K31ac (subsection "H4K31ac marks euchromatin in mammalian cells and apicomplexan parasites" and "GCN5b and HDAC3 enzymes fine-tune the H4K31ac levels in Toxoplasma gondii").

6) The presentation of genomics data relies too heavily on genome browser snapshots. These are helpful and appropriate as supporting evidence, but conclusions based on genomic data should be supported by genomic analyses. Some conclusions, like the limitation of H4K31me1 by translational initiation and termination sites (subsection "Nucleosomes with H4K31ac and H4K31me1 are enriched at the promoter and the core body of active genes, respectively") are not supported at all – transcriptional initiation and termination sites could just as easily be involved. The claim that H3K9me3 and H3K14ac co-localize in poorly transcribed (Q4) genes (Figure 8) is inconsistent with the genomic analysis in Figure 7—figure supplement 1, which does not show an obvious enrichment of either modification in Q4 genes. This may be because Q4 genes are transcriptionally heterogeneous (Figure 7).

After re-reading the manuscript with the reviewers' comments in mind. While you did not require us to do so, we have re-run our analyses to include new data that overall reinforce our initial hypothesis. Indeed, we assessed a variety of RNA-seq expression data (available at ToxoDB and published by Hehl et al., 2015 and Pittman et al., 2014) to re-evaluate the enrichment of H3K9me3 and H3K14ac at genes clustered in Q4. First we uncovered that Q4 genes were indeed transcriptionally heterogeneous. As such, we discovered that the repressive signature H3K9me3 was primarily present at the vicinity of genes typified by low RNA levels in tachyzoites and referenced as stage-specific within the *T. gondii* life cycle. Indeed, we have uncovered that bradyzoite-restricted genes (BAG1, ENO1, SRS) displayed marks of both active (H3K14ac) and silent (H3K9me3) chromatin. This dual histone PTM was also detected on nucleosomes surrounding genes that have been recognized as exclusively expressed during sexual stages in the definitive feline host, like the merozoite gene *GRA11b*. While we have shown few examples in the previous version of the manuscript, we are now providing a tremendous amount of data supporting the conclusion that H3K14ac and H3K9me3 form a bivalent chromatin domain capable of silencing developmental genes while keeping them poised for rapid activation upon cell differentiation. For instance, over the 111 members of the SRS superfamily of proteins annotated in ToxoDB, we identified 66 SRS genes displaying H3K9me3/H3K14ac enrichment, including the aforementioned 52 merozoite (Hehl et al., 2015) along with 8 bradyzoite SRS genes (Pittman et al., 2014) whereas the 14 SRS genes exclusively expressed in tachyzoite (e.g. SAG1 cluster) lack the dual PTM. All these data are discussed (second paragraph in subsection "Interplay between H4K31ac and H4K31me1 predicts distinctive patterns of gene expression in T. gondii") and displayed in the main new Figure 10 as well as in three related Figure 10—figure supplement 1,Figure 10—figure supplement 2 and Figure 10—figure supplement 3.

7) Figure 6 and subsection "Nucleosomes with H4K31ac and H4K31me1 are enriched at the promoter and the core body of active genes, respectively" state that H4K31me1 is a novel mark of transcriptional units, but H3K4me1 (genome-wide data available on Toxodb) seems to have a very similar distribution (also suggested by H3K4me1 data in Figure 7—figure supplement 1, though not discussed), so this statement may not be accurate. Did the authors compare H3K4me1 and H3K31me1 distributions?

This reviewer is right. H3K4me1 displayed a similar distribution over the body of genes that H4K31me1 as reported in ToxoDB but still unpublished by Kami Kim lab. We have reexamined the extent and genome-wide scattering of H3K4me1 and observed a high degree of similarity in read coverage between H3K4me1 and H4K31me1 and a similar pattern of enrichment over the gene body. However, those PTMs are unevenly distributed across the genome, H3K4me1 being occasionally weakly enriched relative to H4K31me1 at few expressed genes. These new data have been now included in Figure 8—figure supplement 3 and commented in the text.

8) Figure 8 shows that H4K31me1 marks Tg pericentromeric heterochromatin (discussed in paragraph three of subsection "Distribution of H4K31 modifications across the P. falciparum genome reveals 403 H4K31me1 as a novel pericentromeric PTM"), but this information appears to be based upon the genome-wide data for CenH3 published in Brooks et al. 2013 (also available at www.Toxodb.org). The Brooks paper, which showed the localization of H3K9me2/3 to Tg centromeres should be discussed and cited since the authors examine similarities and differences between Tg and Pf. The Tg ortholog of HP1, which binds H3K9me3, was reported by Gissot et al., 2012 (also not cited), and its localization was reported to centromeres and telomeres.

The Brooks et al., 2013 and Gissot et al., 2012 papers are now properly commented and referenced in the new revised version.

9) The data reported for H3K9me3 in this manuscript agree in centromeric localization but conflict in reporting localization to "poised" genes. The definition of "poised" genes is unclear. A table is provided (without a legend or description of samples), but statistical support for enrichment of this mark on poised genes is required, as is further discussion. What proportion of genes in Q4 were poised and were they developmental stage-specific genes?

In the revision, we attempted to describe differences between our data and Brooks et al., 2013, in a balanced manner as following: "It was previously reported that H3K9me3 typifies centrometric heterochromatin in *T. gondii* (Brooks et al., 2011) but this study conflicts with our data in reporting enrichment of the mark to "poised" stagespecific genes. This discrepancy could be explained by the genetic background of the parasite strain since Brooks and colleagues infected human cells with a type I (RH) strain that lost the ability to develop into mature cysts while we used infections with a type II strain (Pru) which is more relevant as it does readily develop tissue cysts and latent infections in animals."

Concerning the proportion of genes clustered in Q4 harboring a poised signature, please see our response to comment # 6 above.

10) There are several species in which ncRNA contribute to the organization of centromeres (although probably not Tg based on the authors' data). The Scherf laboratory has characterized ncRNA in Plasmodium, including several that are transcribed at var loci. Are the H4K31me1 marks that flank centromeres in Pf sites of ncRNA transcription? Does the presence of ncRNA at var loci correlate with H4K31me1 localization?

We have no clear correlation. Whereas H3K9me3/HP1 is clearly associated with var and also GC elements according to Macs2. However, H4K31me1 enrichment was detected, it was unevenly and at low rates in the vicinity of few var (5 over 64).

11) The discussion of H4K31Ac and H4K31me1 functionality is overbroad. There are multiple functional claims: "a key role in gene regulation", "pivotal role...in pericentromeric heterochromatin", "regulating chromosomal condensation and segregation during mitosis", "H4K31me [...] substantially slows the progress of the RNA polymerase on active genes". This paper presents no direct functional data and hence cannot claim to demonstrate any role for H4K31Ac or H4K31me1. Reasonable inferences can be made based on genomic localization and correlations with other PTMs and gene expression, but it's important to be very clear that functional tests are yet to be done. This includes the title, which makes unsupported functional claims.12) The discussion of H4K31Ac and H4K31me1 functionality is vague and the descriptions of the genomic localization of H4K31Ac and H4K31me1 are very difficult to follow. H4K31me1 may function by preventing acetylation (but not have a direct influence on nucleosomes), inhibiting histone exchange (or otherwise repressing transcription directly or through a reader), or both. The authors seems to favor the last possibility, but don't explain why.13) The similarities and differences between T. gondii and P. falciparum need to be discussed more clearly. More generally, references to evolutionary relationships are confusingly vague. "Higher eukaryote" is an anachronistic term with little meaning; in most cases the authors appear to mean "animals". Chromatin organization in apicomplexa is described as unique, but many fungi have similar distributions of active and silent chromatin. P. falciparum and T. gondii are described as "closely linked" despite a divergence time of about 800 MY.

Answer to comments 11, 12 and 13:

We agree with those assessments. As such, we have endeavored to extend the paper\x92s contributions by enhancing our Discussion section. We now discuss the specific issues the reviewers highlight. We also worked to improve the tone of the paper. Thus, as we revised the paper, we worked to provide a more balanced presentation of our arguments and results. However, we are hesitant to change the present title which is to our opinion illustrates quite well our data. We hope that our revision, based on the input of the reviewers, accomplishes this objective.

[Editors' note: further revisions were requested prior to acceptance, as described below.]

Essential revisions:1) Antibody specificity. The figure supplements for Figure 1 are difficult to interpret. The authors test the specificity of the antibodies using dot blots and modified histone peptide arrays. In these figures, part C shows a bar graph of a "specificity factor". The relevant figure legends direct the reader to the Materials and methods section, but we could not find an explanation of what a specificity factor is or how it is calculated in the Methods, and more generally the Materials and methods section does not explain how antibody specificity was determined. This needs to be explained so that the readers can understand whether the specificity factors for the H4 K31Ac antibody for modifications such as H3 T11P are significant. Part D of these figures presumably shows the quantitation of the binding of the antibody to the peptide arrays. However, again there is no explanation of how this data was derived or even what the Y-axis represents. This needs to be explained.

We have indeed forgotten to include the material and method. This is now done in the new revised version. The methods are now clearly explained including the significance of the specificity factor.

Based on the data shown, it appears that the H4 K31 Ac antibody can bind to the unmodified H3 tail. If this is the case, the immunofluorescence data in the manuscript is impossible to interpret. Based on the tight cropping of the gels in Figure 1 it is not possible to tell whether the H4 K31Ac antibody recognizes H3. The full gels should be provided.

We observed a nonsignificant specificity value (below 10) of H4K31ac antibody toward H3 modified peptides (Figure 1—figure supplement 1 Part C) to put in perspective with the high values for H3K14ac or H4K20me3 (ranging from 4,000 to 12,000) typifying the specific recognition of the modified peptides. We therefore considered the cross-reactivity against the unmodified H3 tail as non-significant. To support this observation, H4K31ac did not recognize H3 by western blot. We are providing below as requested the full gel corresponding to the H4K31ac of Figure 1 which unquestionably shows no reactivity of the antibody with histone H3.

2) Unwarranted mechanistic claims. The manuscript still contains mechanistic claims about the function of modifications on H4 K31 that are speculative and not supported by data. The primary example of this is in the Abstract where the authors state "H4K31 acetylation promotes a relaxed chromatin state at the promoter of active genes through nucleosome disassembly in both parasites." The manuscript shows that H4 K31 Ac localization correlates with active chromatin but there is no data that shows that H4 K31 acetylation has a direct effect on chromatin structure through nucleosome disassembly. These types of claims should be removed.

As suggested, we replaced in the Abstract the aforementioned sentence by the following: "H4K31 acetylation at the promoter correlates with, and perhaps directly regulates, gene expression in both parasites". We have also removed when needed unwarranted mechanistic claims throughout the new revised version.